# Discriminative Finetuning of Generative Large Language Models without Reward Models and Human Preference Data

**Siqi Guo** [1]   **Ilgee Hong** [2][†]   **Vicente Balmaseda** [1]   **Changlong Yu** [3]   **Liang Qiu** [3]   **Xin Liu** [3]   **Haoming Jiang** [3]
**Tuo Zhao** [3]   **Tianbao Yang** [1]

## Abstract

Supervised fine-tuning (SFT) has become a crucial step for aligning pretrained large language models (LLMs) using supervised datasets of input-output pairs. However, despite being supervised, SFT is inherently limited by its generative training objective. To address its limitations, the existing common strategy is to follow SFT with a separate phase of preference optimization (PO), which relies on either human-labeled preference data or a strong reward model to guide the learning process. In this paper, we address the limitations of SFT by exploring one of the most successful techniques in conventional supervised learning: discriminative learning. We introduce **Discriminative Fine-Tuning (DFT)**, an improved variant of SFT, which mitigates the burden of collecting human-labeled preference data or training strong reward models. Unlike SFT that employs a generative approach and overlooks negative data, DFT adopts a **discriminative paradigm** that increases the probability of positive answers while suppressing potentially negative ones, aiming for **data prediction** instead of token prediction. Our contributions include: (i) a discriminative probabilistic framework for fine-tuning LLMs by explicitly modeling the discriminative likelihood of an answer among all possible outputs given an input; (ii) efficient algorithms to optimize this discriminative likelihood; and (iii) extensive experiments demonstrating DFT's effectiveness, achieving performance better than SFT and comparable to if not better than SFT→PO. The code can be found

at https://github.com/Optimization-AI/DFT.

## 1 Introduction

Fine-tuning large language models (LLMs) has become an essential step to adapt pretrained models to specific tasks, significantly improving their performance and practical utility (Wei et al., 2022; Ouyang et al., 2022; Touvron et al., 2023; OpenAI, 2024; Guo et al., 2025). While pretraining enables LLMs to acquire vast amounts of general knowledge, fine-tuning tailors the model to exhibit desirable behaviors, and excel in specialized domains. As LLMs become integral to applications like conversational AI, content generation, and decision-making, developing effective and efficient fine-tuning methods remains a critical challenge.

The current standard for aligning LLMs typically involves supervised fine-tuning (SFT) followed by preference optimization (PO) denoted by SFT→PO, including techniques such as reinforcement learning from human feedback (RLHF) (Stiennon et al., 2020; Ouyang et al., 2022; Bai et al., 2022; Rafailov et al., 2024). In this approach, SFT first aligns the model with supervised data, and PO further refines the model using preference data labeled by humans or a reward model that simulates human preferences. This two-stage process has achieved significant success, particularly in improving human alignment and response quality. However, PO methods often require extensive human annotations or the construction of robust reward models, both of which are resource-intensive and may limit scalability and applicability in highly specialized areas.

This raises an intriguing question: *Can we align LLMs without human preference data or reward models while achieving competitive performance to SFT→PO?*

To address this question, we propose **D**iscriminative **F**ine-**T**uning (DFT), a novel alternative to SFT→PO that mitigates the burden of collecting human-labeled preference data or training strong reward models. Unlike SFT, which uses a generative approach and overlooks negative examples, DFT adopts a discriminative notion, explicitly discriminating good from "bad" outputs generated by the base model to be finetuned. We formalize this approach by introducing

[1]Department of Computer Science and Engineering, Texas A&M University, College Station, USA [2]H. Milton Stewart School of Industrial and Systems Engineering, Georgia Institute of Technology, Atlanta, USA [3]Amazon.com Inc, CA, USA. Correspondence to: Tianbao Yang <tianbao-yang@tamu.edu>. [†]I.H.'s work was done during the internship at Amazon.

*Proceedings of the 42$^{nd}$ International Conference on Machine Learning*, Vancouver, Canada. PMLR 267, 2025. Copyright 2025 by the author(s).

a discriminative probabilistic framework that models the **discriminative likelihood** of an answer among all possible outputs for a given input. This stands in stark contrast to SFT, which uses a generative probabilistic framework to model only the generative likelihood of individual tokens in the answers. To implement this framework, we develop efficient algorithms to optimize the discriminative likelihood of good answers, ensuring both scalability and practicality. Due to its strong discriminative capability, DFT delivers competitive or superior performance compared to SFT→PO, mitigating the requirement for human preference data and reward models.

Our main contributions are summarized as follows:

- **A novel discriminative framework:** We introduce a probabilistic framework that explicitly models the discriminative likelihood of an answer among all possible outputs, in contrast to the generative likelihood approach used in SFT.

- **Efficient optimization algorithms:** We propose scalable and practical methods to maximize the discriminative likelihood of good answers, ensuring effective fine-tuning of LLMs.

- **Extensive empirical validation:** We conduct extensive experiments to demonstrate that DFT consistently outperforms standard SFT and achieves comparable results than preference optimization methods that rely on explicit preference datasets.

These contributions establish DFT as a new paradigm for enhancing pretrained language models, offering both theoretical and practical advancements in the field.

## 2 Related Work

**Supervised Finetuning.** The standard approach to SFT is mimicking pretraining, which maximizes the likelihood of tokens in the output response given input prompt (Ouyang et al., 2022; Wei et al., 2022; Xu et al., 2023a; Wang et al., 2023; Zhang et al., 2023; Li et al., 2024b). Although simple to implement, it often captures only superficial patterns rather than fostering a deeper understanding of task semantics (Kung & Peng, 2023; Zhang et al., 2023; Gudibande et al., 2024). Recent works have studied inverse reinforcement learning to address this limitation, but such methods often involve interdependent updates between different models (e.g., the policy and a reward-related model), which complicates the training process (Li et al., 2024a; Wulfmeier et al., 2024). In contrast, our method neither requires online sampling from the current policy model nor involves a reward-related model, making it much more efficient.

**Preference Optimization.** Pioneering works proposed RL-based PO methods (Christiano et al., 2017; Ziegler et al.,

2019; Stiennon et al., 2020; Ouyang et al., 2022; Bai et al., 2022). These methods leverage a separate reward model, trained on human-labeled preference data, and optimize the SFT model against it using policy gradient methods such as PPO (Schulman et al., 2017) and REINFORCE (Williams, 1992). Rafailov et al. (2024) proposed direct preference optimization (DPO), which removes the step of training a reward model and directly optimizes a pairwise loss of the policy model on the preference data. Following DPO, many PO methods have been proposed with different loss functions, including R-DPO (Park et al., 2024), CPO (Xu et al., 2024a), IPO (Azar et al., 2024), SimPO (Meng et al., 2024), KTO (Ethayarajh et al., 2024), ORPO (Hong et al., 2024), DPO-p (Pal et al., 2024), to name just a few among others (Zhao et al., 2023; Jung et al., 2024).

Several works (Chen et al., 2024b; Yuan et al., 2023a; Song et al., 2024) have considered PO with a list of ranked preference data that may be explicitly labeled with a reward value. Rosset et al. (2024) assumed a general preference function is given that can produce a probability telling one output is preferred over another output given an input. Different from these works, we do not assume any preference data or preference model other than annotated input-output pairs.

**Finetuning via Self-play.** Training a model on its own self-generated responses has been widely explored in the PO stage. For example, many variants of RLHF (Christiano et al., 2017; Ziegler et al., 2019; Stiennon et al., 2020; Ouyang et al., 2022; Bai et al., 2022; Li et al., 2023; Chan et al., 2024; Ji et al., 2024) use on-policy samples produced by the current policy under optimization. Some studies have exhibited benefits of using self-generated data for PO by reducing the distribution gap between the training data and the current model while fostering exploration of diverse response spaces (Xu et al., 2024b; Tajwar et al., 2024; Tang et al., 2024). Moreover, leveraging synthetic data has proven essential for iterative (online) algorithmic improvement of these methods (Xu et al., 2023b; Guo et al., 2024; Yuan et al., 2024; Chen et al., 2024a; Dong et al., 2024). A more closely related work is SPIN (Chen et al., 2024c), which uses a similar preference optimization objective as DPO but with data generated by the model to be finetuned as the losing responses. Although we also use self-generated data from the base model to be finetuned as our negative data, our formulation is derived from a discriminative learning framework, making our approach aided by advanced optimization better than the pairwise loss function used in SPIN and other pairwise preference optimization objectives (cf. Section 6).

## 3 Preliminaries: SFT

For SFT, we are given a set of data $\mathcal{D} = \{(\mathbf{x}_i, \mathbf{y}_i), i = 1, \ldots, n\}$, where $\mathbf{x}_i$ is an input prompt and $\mathbf{y}_i$ is a labeled output answer. Both the input $\mathbf{x}$ and output $\mathbf{y}$ are expressed

as a sequence of tokens from a vocabulary of size $K$ denoted by $\mathcal{V} = \{v_1, \ldots, v_K\}$. We let $\mathbf{x} = (x_{[1]}, \ldots, x_{[k]})$ and $\mathbf{y} = (y_{[1]}, \ldots, y_{[m']})$, where $x_{[i]} \in \mathcal{V}, y_{[j]} \in \mathcal{V}, \forall i, j$.

SFT considers the next-token prediction in the output $\mathbf{y}$ given an input $\mathbf{x}$. For an input-output pair $(\mathbf{x}, \mathbf{y})$, it models the conditional probability of $\mathbf{y}$ given $\mathbf{x}$ by $P_g(\mathbf{y}|\mathbf{x}) = \prod_{j=1}^{m'} p_g(y_j|\mathbf{x}, y_1, \ldots, y_{j-1})$. The token conditional probability $p_g(x_j|x_1, \ldots, x_{j-1})$ is modeled by a Transformer:

$$p_g(x_j|x_1, \ldots, x_{j-1})$$
$$= \frac{\exp(h(\mathbf{w}; x_1, \ldots, x_{j-1})^\top W_{x_j})}{\sum_{k=1}^K \exp(h(\mathbf{w}; x_1, \ldots, x_{j-1})^\top W_k)},$$

where $W_1, \ldots, W_K$ denotes the token embedding vectors of that in $\mathcal{V}$, $h(\mathbf{w}; x_1, \ldots, x_{j-1})$ denotes the representation of the input sequence of tokens produced by a transformer network parameterized by $\mathbf{w}$. We let $\theta = (\mathbf{w}, W)$ to denote all parameters of the LLM.

By minimizing the negative log-likelihood of all $\mathbf{y}_1, \ldots, \mathbf{y}_n$, SFT solves the following problem from a pretrained model:

$$\min_\theta -\frac{1}{n} \sum_{i=1}^n \log P_g(\mathbf{y}_i|\mathbf{x}_i). \quad (1)$$

In order to differentiate from our approach, we refer to $P_g(\mathbf{y}|\mathbf{x})$ as the **generative likelihood**, as it decomposes the likelihood of generating $\mathbf{y}$ given $\mathbf{x}$ into the product of likelihood of generating each token in $\mathbf{y}$.

## 4 DFT: Discriminative Finetuning

In order to motivate our approach, let us first examine the limitation of SFT. Our goal of finetuning LLMs is to ensure that LLMs generate good answers more likely than bad answers. However, SFT only has one-sided optimization power by maximizing the likelihood of tokens in the good output $\mathbf{y}$ given $\mathbf{x}$ and their preceding tokens. It does not necessarily guarantee that the likelihood of tokens in the bad answer is low. Let us consider a simple example:

> **Motivation Example**
>
> $(\mathbf{x})$ What is the bigger number between 9.11 and 9.9?
> $(\mathbf{y})$ The bigger number between 9.11 and 9.9 is 9.9.
> $(\mathbf{y}')$ The bigger number between 9.11 and 9.9 is 9.11.

The good answer $\mathbf{y}$ and the bad answer $\mathbf{y}'$ only differ in the last token. The likelihood of all preceding tokens are the same. Even though the likelihood of the last token "9" in $\mathbf{y}$ conditioned on preceding tokens is increased during the finetuning with this data, the likelihood of the token "11" as the last one might still be high, making generating the bad answer $\mathbf{y}'$ likely.

To address this issue, the current mainstream approach is to finetune the model further using PO on human preference

data. If humans label the two answers such that $\mathbf{y} \succ \mathbf{y}'$, the model might be able to push the likelihood of $\mathbf{y}'$ given $\mathbf{x}$ smaller than that of $\mathbf{y}$ given $\mathbf{x}$. As a result, the good answer $\mathbf{y}$ will be generated more likely than the bad answer $\mathbf{y}'$.

However, traditional supervised learning methods never use human preference data. For example, in image classification, training data $(\mathbf{x}, y)$ denote an input image and its true class label $y \in \{1, \ldots, K\}$. We do not need the preference optimization step on preference data saying that a dog class is preferred to a cat class for an image of a dog. So what is the difference between traditional supervised learning and supervised finetuning of LLMs that makes SFT not enough? The answer lies in the fact that *traditional supervised learning methods are usually* **discriminative approaches***, while the standard SFT method is not discriminative.*

Below, we introduce our discriminative finetuning (DFT) framework of LLMs. A discriminative approach aims to push the "score" of the true output to be higher than that of other possibly wrong outputs. In this paper, we examine a classical approach through **discriminative probabilistic model.** To this end, we introduce a parameterized scoring function $s_\theta(\mathbf{y}, \mathbf{x}) \in \mathbb{R}$, which measures the fitness of $\mathbf{y}$ given $\mathbf{x}$. This is similar to the prediction score in traditional supervised learning. We will discuss shortly how to set the scoring function for learning an LLM. In a discriminative probabilistic model, we model the conditional probability $P_d(\mathbf{y}|\mathbf{x})$ of one output $\mathbf{y}$ out of the space of all possible texts denoted by $\mathcal{Y}$. In particular, we define

$$P_d(\mathbf{y}|\mathbf{x}) = \frac{\exp(s_\theta(\mathbf{y}, \mathbf{x})/\tau)}{\sum_{\mathbf{y}' \in \mathcal{Y}} \exp(s_\theta(\mathbf{y}', \mathbf{x})/\tau)}, \forall \mathbf{y} \in \mathcal{Y}, \quad (2)$$

where $\tau > 0$ is a temperature hyperparameter. Then, given a set of training data $\mathcal{D} = \{(\mathbf{x}_1, \mathbf{y}_1), \ldots, (\mathbf{x}_n, \mathbf{y}_n)\}$, we learn $\theta$ by maximizing the log-likelihood of observed data, i.e.,

$$\min_\theta \quad F(\theta) \quad (3)$$

where $F(\theta) := -\frac{1}{n} \sum_{i=1}^n \tau \log P_d(\mathbf{y}_i|\mathbf{x}_i) =$

$$-\frac{1}{n} \sum_{i=1}^n s_\theta(\mathbf{y}_i, \mathbf{x}_i) + \frac{\tau}{n} \sum_{i=1}^n \log \left[ \sum_{\mathbf{y}' \in \mathcal{Y}} \exp\left(\frac{s_\theta(\mathbf{y}', \mathbf{x}_i)}{\tau}\right) \right],$$

where scaling the negative log-likelihood by $\tau$ is for increasing the numerical stability, which does not change the optimal solution.

DFT marks a paradigm shift from "token" prediction to "data" prediction. To differentiate from the generative likelihood $P_g(\mathbf{y}|\mathbf{x})$, we refer to $P_d(\mathbf{y}|\mathbf{x})$ in (2) as **discriminative likelihood** of $\mathbf{y}$ given $\mathbf{x}$. By maximizing the discriminative log-likelihood of the training data, we not only increase the score of the true output $\mathbf{y}_i$ for each input $\mathbf{x}_i$, corresponding to the numerator of the discriminative likelihood, but also decrease the scores of other potentially bad answers in $\mathcal{Y}$,

which correspond to the denominator of the discriminative likelihood.

Finally, we note that while both DFT and traditional discriminative classification methods (e.g., logistic regression) use supervised data $\mathcal{D} = \{(\mathbf{x}_i, \mathbf{y}_i), i = 1, \ldots, n\}$ and model discriminative likelihood through softmax functions, the differences are: (1) DFT operates over an infinite space of possible text outputs $\mathbf{y}'$, whereas traditional classification works with a finite set of class labels; (2) the summation over all possible $\mathbf{y}'$ in DFT requires advanced optimization techniques, while summation over class labels in traditional approaches is straightforward. This fundamental distinction necessitates our novel sampling and estimation approach while maintaining the core advantages of discriminative learning principles.

### 4.1 The Scoring Function

The above framework is similar to the discriminative probabilistic modeling for self-supervised representation learning (Wang et al., 2025). However, we cannot directly borrow the same idea of discriminative representation learning to design the scoring function. In particular, discriminative representation learning uses an encoder network $e(\mathbf{x})$ to induce an embedding of any input text $\mathbf{x}$, and computes the scoring function by using the cosine similarity between $e(\mathbf{x})$ and $e(\mathbf{y})$. However, this representation model $e(\cdot)$ is of no use for generative tasks of LLMs.

To circumvent this issue, we define the scoring function based on the generative log-likelihood $\log P_g(\mathbf{y}|\mathbf{x})$, as it measures the likeliness of generating $\mathbf{y}$ given $\mathbf{x}$. For a good model, we expect that a high value of the generative log-likelihood $\log P_g(\mathbf{y}|\mathbf{x})$ would indicate a high fitness score of $\mathbf{y}$ to answer $\mathbf{x}$. With such correspondence, the above discriminative learning framework would increase the chance of generating a good output $\mathbf{y}$ given $\mathbf{x}$ and decrease the chance of generating possibly bad outputs given $\mathbf{x}$. We will examine two simple settings of the scoring function.

**Setting 1:** $s_\theta(\mathbf{y}, \mathbf{x}) = \log P_g(\mathbf{y}|\mathbf{x})$. Plugging this into (3) results in the following objective:

$$\min_\theta -\frac{1}{n}\sum_{i=1}^n \log P_g(\mathbf{y}_i|\mathbf{x}_i)$$
$$+ \tau \frac{1}{n}\sum_{i=1}^n \log\left(\sum_{\mathbf{y}'\in\mathcal{Y}} \exp\left(\frac{\log P_g(\mathbf{y}'|\mathbf{x}_i)}{\tau}\right)\right). \quad (4)$$

Comparing the above objective of DFT to that of SFT in (1), we can see that the first term in (4) is exactly the same as the objective of SFT. The difference lies in the second term that penalizes the possibly bad outputs in $\mathcal{Y}$ for each $\mathbf{x}_i$, trying to decrease their generative log-likelihood.

**Setting 2:** For the second setting, we use length normalized generative log-likelihood as the scoring function, e.g.,

$s_\theta(\mathbf{y}, \mathbf{x}) = \frac{1}{|\mathbf{y}|}\log P_g(\mathbf{y}|\mathbf{x})$, where $|\mathbf{y}|$ denotes the number of tokens in $\mathbf{y}$. This will allow us to compare DFT with some PO approaches using length normalized reward (Meng et al., 2024). As a result, the problem becomes:

$$\min_\theta -\frac{1}{n}\sum_{i=1}^n \frac{1}{|\mathbf{y}_i|}\log P_g(\mathbf{y}_i|\mathbf{x}_i)$$
$$+ \tau\frac{1}{n}\sum_{i=1}^n \log\left(\sum_{\mathbf{y}'\in\mathcal{Y}} \exp\left(\frac{\log P_g(\mathbf{y}'|\mathbf{x}_i)}{|\mathbf{y}'|\tau}\right)\right). \quad (5)$$

### 4.2 The Optimization Algorithm

Although our DFT formulations (2) and (3) are nearly identical to that of the traditional discriminative probabilistic approach for classification (e.g., logistic regression), the key challenge lies in solving the optimization problem in (3), particularly in handling the second term of $F(\theta)$, where $\mathcal{Y}$ encompasses all possible texts. Indeed, the optimization problem in (3) is an instance of **empirical X-risk minimization** (Yang, 2022; Yuan et al., 2023b). We address the optimization challenge by employing advanced optimization techniques of **finite-sum coupled compositional optimization framework (FCCO)** (Wang & Yang, 2022). The idea is to write the second term of $F(\theta)$ into the form of $\frac{1}{n}\sum_{i=1}^n f(\mathbb{E}_\zeta g_i(\theta; \zeta))$, where $\zeta$ is some random variable. To this end, we introduce a sampling distribution $P_i(\cdot)$, which is specified later. Then we define

$$g_i(\theta) := \sum_{\mathbf{y}'\in\mathcal{Y}} \exp\left(\frac{s_\theta(\mathbf{y}', \mathbf{x}_i)}{\tau}\right) = \mathbb{E}_{\mathbf{y}'\sim P_i(\cdot)} \frac{\exp\left(\frac{s_\theta(\mathbf{y}', \mathbf{x}_i)}{\tau}\right)}{P_i(\mathbf{y}')}.$$

The objective becomes:

$$\min_\theta -\frac{1}{n}\sum_{i=1}^n s_\theta(\mathbf{y}_i, \mathbf{x}_i)$$
$$+ \frac{1}{n}\sum_{i=1}^n \tau\log\left(\mathbb{E}_{\mathbf{y}'\sim P_i(\cdot)} \frac{\exp(s_\theta(\mathbf{y}', \mathbf{x}_i)/\tau)}{P_i(\mathbf{y}')}\right). \quad (6)$$

Next, we discuss three components of our algorithm for solving the above problem.

**Sampling Distributions.** We need three properties of these sampling distributions: (1) it is easy to sample data from them; (2) it is possible to compute the probability value of a sample $\mathbf{y}'$; (3) the sampled outputs $\mathbf{y}' \sim P_i(\cdot)$ are likely to be bad outputs in answering $\mathbf{x}_i$. To this end, we let $P_i(\cdot) = P_g^0(\cdot|\bar{\mathbf{x}}_i)$, where $P_g^0$ corresponds to the base LLM $\theta_0$ to be finetuned, and $\bar{\mathbf{x}}_i$ is an augmented text of $\mathbf{x}_i$ including some system prompts to facilitate the generation of bad outputs. We explore this in our experiments.

**Key Updates.** Computing a stochastic gradient estimator for the first term is the same as in SFT. The challenge is how to estimate the gradient of $\tau\log(g_i(\theta_t))$ in the second term using random samples. Its gradient is given by $\frac{\tau}{g_i(\theta_t)}\nabla g_i(\theta_t)$. Although $\nabla g_i(\theta_t)$ can be simply estimated

**Algorithm 1** The DFT Algorithm

1: Initialize $\theta_0$ as the base LLM, and $\mathbf{u}_0 = \mathbf{1}$
2: **for** $t = 0, 1, \ldots, T - 1$ **do**
3:     Sample a mini-batch $\mathcal{S}_t \subset \{\mathbf{x}_1, \ldots, \mathbf{x}_n\}$
4:     **for** each $\mathbf{x}_i \in \mathcal{S}_t$ **do**
5:        Sample a mini-batch $\mathcal{B}_{i,t}^0$ from $P_g^0(\cdot|\bar{\mathbf{x}}_i)$ via an offline pool
6:        Update $u_{i,t+1}$ according to (7)
7:     **end for**
8:     Compute a gradient estimator $G_t$ according to (8)
9:     Update $\theta_{t+1}$ using Adam-W
10: **end for**

using an unbiased stochastic gradient, estimating $\frac{\tau}{g_i(\theta_t)}$ cannot simply use an unbiased stochastic estimator of $g_i(\theta_t)$ since it is a non-linear function $g_i(\theta_t)$, which will yield a biased estimator. Following Wang & Yang (2022), we maintain and update $n$ moving average estimators $\{u_1, \ldots, u_n\}$ to track $g_i(\theta)$ for each $\mathbf{x}_i$. In particular, at the $t$-th iteration given a solution $\theta_t$, we first sample a mini-batch of data $\mathcal{S}_t \subset \{\mathbf{x}_1, \ldots, \mathbf{x}_n\}$. For each data $\mathbf{x}_i \in \mathcal{S}_t$, we sample one or multiple outputs $\mathbf{y}' \sim P_g^0(\cdot|\bar{\mathbf{x}}_i)$, e.g., by generating them through feeding $\bar{\mathbf{x}}_i$ as the input prompt to the base LLM $P_g^0$. We denote these outputs as $\mathcal{B}_{i,t}^0 = \{\mathbf{y}_{i,t,1}', \ldots, \mathbf{y}_{i,t,B}'\}$. Then we update $u_{i,t+1}$ by:

$$u_{i,t+1} = (1 - \gamma)u_{i,t} + \gamma \frac{1}{B} \sum_{\mathbf{y}' \in \mathcal{B}_{i,t}^0} \frac{\exp(\frac{s_{\theta_t}(\mathbf{y}',\mathbf{x}_i)}{\tau})}{P_g^0(\mathbf{y}'|\bar{\mathbf{x}}_i)}, \quad (7)$$

where $\gamma \in (0, 1)$. With $u_{i,t}$, the gradient of $\tau \log(g_i(\theta_t))$ can be estimated by $\frac{\tau}{u_{i,t+1}}\nabla\widehat{g}_i(\theta_t)$, where

$$\nabla\widehat{g}_i(\theta_t) = \frac{1}{B} \sum_{\mathbf{y}' \in \mathcal{B}_{i,t}^0} \frac{\exp(\frac{s_{\theta_t}(\mathbf{y}',\mathbf{x}_i)}{\tau})\nabla s_{\theta_t}(\mathbf{y}',\mathbf{x}_i)}{\tau P_g^0(\mathbf{y}'|\bar{\mathbf{x}}_i)}$$

denotes a mini-batch estimator of $\nabla g_i(\theta_t)$. We emphasize that the moving average estimator (7) is critical to calculating an accurate gradient estimator of the objective. In our experiments, we show $\gamma = 1$ (i.e., simply using a mini-batch estimator of $g_i(\theta_t)$) will yield much worse performance.

Thus, we compute an estimator of the gradient $\nabla F(\theta_t)$ by:

$$G_t = -\frac{1}{|\mathcal{S}_t|} \sum_{\mathbf{x}_i \in \mathcal{S}_t} \nabla s_{\theta_t}(\mathbf{y}_i, \mathbf{x}_i) +$$

$$\frac{1}{|\mathcal{S}_t|} \sum_{\mathbf{x}_i \in \mathcal{S}_t} \frac{1}{u_{i,t+1}B} \sum_{\mathbf{y}' \in \mathcal{B}_{i,t}^0} \frac{\exp(\frac{s_{\theta_t}(\mathbf{y}',\mathbf{x}_i)}{\tau})\nabla s_{\theta_t}(\mathbf{y}',\mathbf{x}_i)}{P_g^0(\mathbf{y}'|\bar{\mathbf{x}}_i)}. \quad (8)$$

Finally, we can update the model parameter $\theta_{t+1}$ following the momentum-based methods (e.g., Adam, Adam-W). This method has a provable convergence guarantee for solving (3) following Wang & Yang (2022). Our optimization method is summarized in Algorithm 1.

**Efficient Implementation.** There are several implementa-

tion issues of Algorithm 1 that are worth discussing.

The first issue is the Step 5, which sample outputs from the sampling model $P_g^0(\cdot|\bar{\mathbf{x}}_i)$. This could increase the training time if it is done online. However, since the sampling model is fixed, we can generate these data offline for all $\mathbf{x}_i, i = 1, \ldots, n$. This could dramatically reduce our training time. In our experiments, we generate $m = E \times B$ number of outputs for each data $\mathbf{x}_i$ from the sampling model and sample $B$ outputs from this pool in Step 5 without replacement, where $E$ is the number of epochs.

Another issue is the numerical stability when calculating the stochastic gradient estimator in Step 8 (c.f. (8)). Take $s_\theta(\mathbf{y}, \mathbf{x}) = \log P_g(\mathbf{y}|\mathbf{x})$ as an example. Then, $\exp(s_{\theta_t}(\mathbf{y}', \mathbf{x}_i)/\tau) = P_g(\mathbf{y}'|\mathbf{x})^{1/\tau}$, which could be a very small value as we are trying to decrease the generative likelihood of generated outputs $\mathbf{y}'$. As a result, the value of estimators $u_{i,t}$ can be extremely small, e.g., $10^{-x}$ where $x$ is a large number, causing some numerical issues. This issue is tackled by maintaining and updating $\{\log u_1, \ldots, \log u_n\}$ instead of $\{u_1, \ldots, u_n\}$. Specifically, we denote by $w_{i,t,\mathbf{y}'} = \frac{\exp(\frac{s_{\theta_t}(\mathbf{y}',\mathbf{x}_i)}{\tau})}{P_g^0(\mathbf{y}'|\bar{\mathbf{x}}_i)}$. Then (7) can be reformulated to:

$$\exp(\log u_{i,t+1}) = \exp(\log(1 - \gamma) + \log u_{i,t})$$
$$+ \exp(\log \gamma + \log \frac{1}{B} \sum_{\mathbf{y}' \in \mathcal{B}_{i,t}^0} w_{i,t,\mathbf{y}'}).$$

For simplicity, let $b_{i,t} = \log(1 - \gamma) + \log u_{i,t}$ and $w_{i,t} = \log \gamma + \log \frac{1}{B} \sum_{\mathbf{y}' \in \mathcal{B}_{i,t}^0} w_{i,t,\mathbf{y}'}$, we have

$$\exp(\log u_{i,t+1}) = \exp(b_{i,t}) + \exp(w_{i,t})$$

If $w_{i,t} < b_{i,t}$, we let

$$\exp(\log u_{i,t+1}) = \exp(b_{i,t})(1 + \exp(w_{i,t} - b_{i,t}));$$

otherwise, we let

$$\exp(\log u_{i,t+1}) = \exp(w_{i,t})(1 + \exp(b_{i,t} - w_{i,t})).$$

Combining these two cases, we have the following:

$$\exp(\log u_{i,t+1}) \quad (9)$$
$$= \exp(\max\{b_{i,t}, w_{i,t}\})(1 + \exp(-|b_{i,t} - w_{i,t}|))$$
$$= \exp(\max\{b_{i,t}, w_{i,t}\})\sigma^{-1}(|b_{i,t} - w_{i,t}|),$$

where $\sigma(\cdot)$ denotes the sigmoid function. Taking the log on both sides gives the update for $\log u_{i,t+1}$. To summarize, we maintain and update $\bar{u}_{i,t} = \log u_{i,t}$ as following:

$$b_{i,t} = \log(1 - \gamma) + \bar{u}_{i,t}$$

$$w_{i,t} = \log \gamma + \log \frac{1}{B} \sum_{\mathbf{y}' \in \mathcal{B}_{i,t}^0} w_{i,t,\mathbf{y}'} \quad (10)$$

$$\bar{u}_{i,t+1} = \max\{b_{i,t}, w_{i,t}\} - \log \sigma(|b_{i,t} - w_{i,t}|).$$

Then $G_t$ is calculated using $\exp(\bar{u}_{i,t+1})$ in place of $u_{i,t+1}$.

**Algorithm 2** The DFT2 Algorithm

1: Initialize $\theta_0$ as the base LLM, and $\mathbf{u}_0 = \mathbf{1}$
2: **for** $t = 0, 1, \ldots, T - 1$ **do**
3:    Sample a mini-batch batch $\mathcal{S}_t \subset \{\mathbf{x}_1, \ldots, \mathbf{x}_n\}$
4:    **for** each $\mathbf{x}_i \in \mathcal{S}_t$ **do**
5:       Sample a mini-batch $\mathcal{B}_{i,t}^0$ from $\mathcal{S}_i^0$
6:       Update $\bar{u}_{i,t+1}$ according to (12) and (10)
7:    **end for**
8:    Compute a gradient estimator $G_t$ according to (13)
9:    Update $\theta_{t+1}$ using Adam-W
10: **end for**

## 5 DFT2: An Approximation Approach

Compared with SFT, DFT has an extra cost of computing $P_g^0(\mathbf{y}'|\bar{\mathbf{x}}_i)$ at each forward step during training. We can further reduce this cost by using an approximation. The idea is simple by just using the length-normalized generative log-likelihood as the scoring function $s_\theta(\mathbf{y}, \mathbf{x}) = \frac{1}{|\mathbf{y}|} \log P_g(\mathbf{y}|\mathbf{x})$ and dropping $P_g^0(\mathbf{y}'|\bar{\mathbf{x}}_i)$ in the update of $u_{i,t+1}$ and the gradient estimator $G_t$. Below, we explain this approximation from two perspectives.

We first explain the approximation via approximating $\sum_{\mathbf{y}' \in \mathcal{Y}} \exp(s_\theta(\mathbf{y}', \mathbf{x}))$ by using the data generated by the base LLM model. Let $\mathcal{S}_i^0 = \{\mathbf{y}'_{i,1}, \ldots, \mathbf{y}'_{i,m}\}$ denote a set of outputs sampled for each data $\mathbf{x}_i$ following the base model $P_g^0(\cdot|\bar{\mathbf{x}}_i)$. Considering that the base LLM has already been trained significantly on a large corpus, hence $\exp(s_\theta(\mathbf{y}', \mathbf{x})/\tau)$ for $\mathbf{y}' \sim P_g^0(\cdot|\bar{\mathbf{x}}_i)$ would be much larger than a random $\mathbf{y}'$ in $\mathcal{Y}$. This is verified by Figure 1. Hence, we approximate $\sum_{\mathbf{y}' \in \mathcal{Y}} \exp(s_\theta(\mathbf{y}', \mathbf{x})/\tau) \approx \sum_{\mathbf{y}' \in \mathcal{S}_i^0} \exp(s_\theta(\mathbf{y}', \mathbf{x})/\tau)$. The second explanation is drawn from an observation made in Meng et al. (2024). They observed that the samples from a LLM have roughly the same values of $s_\theta(\mathbf{y}', \mathbf{x}) = \frac{1}{|\mathbf{y}'|} \log P_g(\mathbf{y}'|\mathbf{x})$. Hence, we can approximate $g_i(\theta)$ by

$$g_i(\theta) \approx \frac{1}{m} \sum_{\mathbf{y}' \in \mathcal{S}_i^0} \frac{\exp(\frac{s_\theta(\mathbf{y}', \mathbf{x})}{\tau})}{P_g^0(\mathbf{y}'|\bar{\mathbf{x}}_i)} \propto \frac{1}{m} \sum_{\mathbf{y}' \in \mathcal{S}_i^0} \exp(\frac{s_\theta(\mathbf{y}', \mathbf{x})}{\tau}),$$

where the second step is justified by that $\exp(s_\theta(\mathbf{y}', \mathbf{x})/\tau)$ are approximately the same, hence the weighting by $1/P_g^0(\mathbf{y}'|\bar{\mathbf{x}}_i)$ becomes insignificant.

With either approximation, we end up with the following optimization problem:

$$\min_\theta -\frac{1}{n} \sum_{i=1}^n s_\theta(\mathbf{y}_i, \mathbf{x}_i)$$
$$+ \frac{1}{n} \sum_{i=1}^n \tau \log \left( \frac{1}{m} \sum_{\mathbf{y}' \in \mathcal{S}_i^0} \exp(s_\theta(\mathbf{y}', \mathbf{x})/\tau) \right). \quad (11)$$

We solve the above problem using the same optimization

technique, except for the change on $u_{i,t+1}$ and $G_t$:

$$u_{i,t+1} = (1 - \gamma)u_{i,t} + \frac{\gamma}{B} \sum_{\mathbf{y}' \in \mathcal{B}_{i,t}^0} \exp(\frac{s_{\theta_t}(\mathbf{y}', \mathbf{x}_i)}{\tau}) \quad (12)$$

$$G_t = -\frac{1}{|\mathcal{S}_t|} \sum_{\mathbf{x}_i \in \mathcal{S}_t} \nabla s_{\theta_t}(\mathbf{y}_i, \mathbf{x}_i) + \quad (13)$$

$$\frac{1}{|\mathcal{S}_t|} \sum_{\mathbf{x}_i \in \mathcal{S}_t} \frac{1}{u_{i,t+1}B} \sum_{\mathbf{y}' \in \mathcal{B}_{i,t}^0} \exp(\frac{s_{\theta_t}(\mathbf{y}', \mathbf{x}_i)}{\tau}) \nabla s_{\theta_t}(\mathbf{y}', \mathbf{x}_i).$$

We refer to the algorithm for solving (11) similar to Algorithm 1 with the above updates of $u_{i,t+1}$ and $G_t$ as DFT2.

**Computational Costs:** As shown in Figure 4, DFT2 has a dramatic reduction in computation costs compared with DFT, as it does not need to load the sampling model $P_g^0$ into the memory and compute $P_g^0(\mathbf{y}'|\mathbf{x}_i)$, which DFT requires. Compared with SFT, DFT2 has additional costs for computing $\nabla s_{\theta_t}(\mathbf{y}', \mathbf{x}_i), \mathbf{y}' \in \mathcal{B}_i^0$. Nevertheless, such costs appear in the preference optimization step of existing approaches.

## 6 Comparison with PO and Self-play

Let us compare DFT2 with preference optimization (PO) approaches. A standard setting of PO is to finetune a LLM based on a set of preference data $\{(\mathbf{x}_i, \mathbf{y}_i, \mathbf{y}'_i)\}_{i=1}^n$, where $\mathbf{y}_i$ denotes a winning response to $\mathbf{x}_i$ and $\mathbf{y}'_i$ denotes a losing response, labeled either by a human or a reward model. Most PO approaches can be cast into the following pairwise loss minimization problem:

$$\min_\theta \frac{1}{n} \sum_{i=1}^n \ell(r_\theta(\mathbf{y}_i, \mathbf{x}_i), r_\theta(\mathbf{y}'_i, \mathbf{x}_i)),$$

where $r_\theta(\mathbf{y}, \mathbf{x})$ denotes some reward function. This framework can be easily extended to incorporate multiple losing responses in the preference data $\{(\mathbf{x}_i, \mathbf{y}_i, \mathbf{y}'_{i1}, \mathbf{y}'_{i1}, \ldots, \mathbf{y}'_{im})\}_{i=1}^n$, by solving the following problem:

$$\min_\theta \frac{1}{n} \sum_{i=1}^n \frac{1}{m} \sum_{j=1}^m \ell(r_\theta(\mathbf{y}_i, \mathbf{x}_i), r_\theta(\mathbf{y}'_{ij}, \mathbf{x}_i)). \quad (14)$$

For example, DPO uses a reward function $r_\theta(\mathbf{y}, \mathbf{x}) = \beta \log \frac{P_g(\mathbf{y}|\mathbf{x})}{P_g^0(\mathbf{y}|\mathbf{x})}$ and a logistic loss function $\ell(r_\theta(\mathbf{y}, \mathbf{x}), r_\theta(\mathbf{y}', \mathbf{x})) = -\log \sigma(r_\theta(\mathbf{y}, \mathbf{x}) - r_\theta(\mathbf{y}', \mathbf{x}))$. Self-play finetuning (SPIN) uses the same objective as DPO except for that $\mathbf{y}'$ is generated by the base LLM. SimPO uses a reward function $r_\theta(\mathbf{y}, \mathbf{x}) = \frac{\beta}{|\mathbf{y}|} \log P_g(\mathbf{y}|\mathbf{x})$ and adds a margin parameter $\gamma$ to the logistic loss function $\ell(r_\theta(\mathbf{y}_i, \mathbf{x}_i), r_\theta(\mathbf{y}'_i, \mathbf{x}_i)) = -\log \sigma(r_\theta(\mathbf{y}_i, \mathbf{x}_i) - r_\theta(\mathbf{y}'_i, \mathbf{x}_i) - \gamma)$.

**Similarity:** From the perspective of PO, we can regard the sampled data $\mathbf{y}' \in \mathcal{S}_i^0$ in DFT2 as potentially losing response to $\mathbf{x}_i$ and the given output $\mathbf{y}_i$ as the winning response. Hence, the objective of DFT2 integrates both the

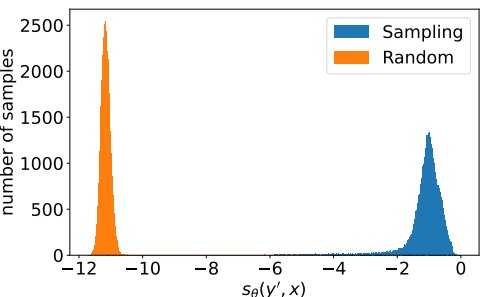

Figure 1: Distribution of $s_\theta(\mathbf{y}', \mathbf{x})$ computed with the model checkpoint after one epoch of training on UltraFeedback data, comparing two groups of $\mathbf{y}'$. Random $\mathbf{y}'$ samples are generated using a very high temperature, while the "Sampling" group represents $\mathbf{y}'$ generated with temperature less than 1 as used in our method.

power of SFT and PO.

**Difference:** To understand the difference between the objective of DFT2 and the pairwise loss used in PO approaches, we can rewrite (11) as:

$$\min_\theta \frac{1}{n} \sum_{i=1}^{n} \tau \log \left( \frac{1}{m} \sum_{\mathbf{y}' \in \mathcal{S}_i^0} \exp\left( \frac{s_\theta(\mathbf{y}', \mathbf{x}) - s_\theta(\mathbf{y}_i, \mathbf{x}_i)}{\tau} \right) \right).$$

It can be seen that the difference between DFT2 and PO is that they use different losses for each data $\mathbf{x}_i$. In particular, DFT2 uses the log-sum-exp loss function $\tau \log(\frac{1}{m} \sum_{\mathbf{y}' \in \mathcal{S}_i^0} \exp(\frac{s_\theta(\mathbf{y}', \mathbf{x}) - s_\theta(\mathbf{y}_i, \mathbf{x}_i)}{\tau}))$ for each data $\mathbf{x}_i$, while PO uses an averaged loss $\frac{1}{m} \sum_{j=1}^{m} \ell(r_\theta(\mathbf{y}_i, \mathbf{x}_i), r_\theta(\mathbf{y}'_i, \mathbf{x}_i))$ for each data. Like the cross-entropy loss for classification or the contrastive loss for self-supervised representation learning, the log-sum-exp loss has the property of giving higher weights to a potentially bad output $\mathbf{y}'$ with a larger score $s_\theta(\mathbf{y}', \mathbf{x})$ in the gradient computation. In contrast, the averaged loss in PO does not enjoy this property.

# 7 Experiments

**Setup**: We evaluate our proposed DFT framework under two distinct training settings. First, we focus on improving the mathematical reasoning capability of a base LLM by using DFT on the MetaMathQA dataset (Yu et al., 2024), which contains 395K samples generated through bootstrapped mathematical questions with augmented reasoning paths. In this setting, we set $B = 4$. Second, we fine-tune a base LLM using DFT on the UltraFeedback (UF) dataset (Cui et al., 2023), comprising 61K samples. UF is originally used as a preference dataset, where each data pairs a winning response $\mathbf{y}_w$ and a losing response $\mathbf{y}_l$. For SFT and DFT, we regard the winning responses $\mathbf{y}_w$ as the ground-truth and discard all losing responses. In this setting, we set $B = 2$ and generate $\mathbf{y}'$ by adding an adversarial prompt like "You

are an unhelpful assistant." to the input $\mathbf{x}_i$ in a chat template (cf. Appendix B.5) and use it as input to the base LLM for generation. For both settings, we use Mistral-7B-v0.1 as our base model. More details of implementation and hyper-parameter tuning are described in Appendix A.

**Evaluation Benchmarks.** For the first training setting, we evaluate our methods on two widely adopted benchmarks: GSM8K (Cobbe et al., 2021) and MATH (Hendrycks et al., 2021b). We use zero-shot test accuracy as our evaluation metric to assess the model's true reasoning capabilities. For the second training setting, we evaluate on seven diverse benchmarks from the Huggingface Open Leaderboard, including MMLU (Hendrycks et al., 2021a), TruthfulQA (Lin et al., 2022), HellaSwag (Zellers et al., 2019), WinoGrande (Sakaguchi et al., 2019), GSM8K (Cobbe et al., 2021), ARC (Clark et al., 2018), and IFEval (Zhou et al., 2023). We follow the few-shot evaluation protocol from Chen et al. (2024c). For IFEval, we report the prompt-level strict accuracy. In addition, we also consider evaluation using GPT4-as-a-judge on AlpacaEval2 (Dubois et al., 2024).

## 7.1 Results

Table 1 shows the performance of DFT(2) on improving mathematical reasoning capabilities. Table 2 and Table 3 compare DFT(2) with self-play methods and SFT→PO methods, respectively, for the second training setting. We describe our observations below.

**Observation 1: DFT variants improve standard SFT.** Both DFT and DFT2 surpass MetaMath-Mistral-7B trained by SFT, achieving state-of-the-art performance among 7B-parameter models on GSM8K (79.15%) and MATH (28.62%). Similarly, for general language tasks (Table 2), DFT improves SFT across almost all benchmarks except for MMLU, on which both methods are competitive. In addition, both DFT and DFT2 outperform SFT on average.

**Observation 2: DFT variants consistently outperform PO methods on self-play data.** In Table 2, we compare DFT(2) with PO approaches using self-play data that is generated by the base model as negative data, including SPIN, SimPO, KTO, ORPO, DPO-p and SimPO-SFT, where the last one just combines the SimPO loss and the SFT loss similar to Xu et al. (2024a). Comparing with SimPO-SFT allows us to verify the advantage of our objective over SimPO loss. For these baselines, we use the same generated $\mathbf{y}'$ as in DFT as their negative data and finetune the same base model. The results in Table 2 show that these PO approaches on self-play data can not improve SFT. This is different from the observation in Chen et al. (2024c); Xu et al. (2024a), as their experiments are for finetuning an SFT model. Comparing DFT(2) with SPIN and SimPO can justify the effectiveness of our objectives and the optimization algorithm.

**Observation 3: DFT is competitive with SFT→PO ap-**

Table 1: Testing accuracy on GSM8K and MATH

| METHOD | GSM8K | MATH |
|---|---|---|
| METAMATH-7B | 66.5 | 19.8 |
| METAMATH-MISTRAL-7B | 77.7 | 28.2 |
| DFT (MISTRAL-7B-BASE) | **79.15** | 28.34 |
| DFT2 (MISTRAL-7B-BASE) | 78.77 | **28.62** |

**proaches.** We compare with state-of-the-art PO methods including DPO, SimPO, RRHF, R-DPO, CPO, IPO, and KTO after SFT in Table 3. It is notable that the existing training pipeline for PO-based methods first trains an SFT model on the UltraChat-200k (UC) dataset (Ding et al., 2023) and then applies PO on the UF preference dataset. However, this training pipeline gives worse performance than SFT on UF winning data. It is probably because the winning responses in the UF dataset give better ground-truth than the data in UC. To mitigate this issue, we implement three methods SFT→DPO, SFT→SimPO and SFT→KTO using an improved pipeline (UF→UF) that first performs SFT on UF winning data and then applies PO on the UF preference data.

Our results in Table 3 show that DFT variants significantly outperform all methods using the standard UC→UF pipeline. When compared to the improved UF→UF pipeline, DFT enjoys competitive performance with state-of-the-art PO methods including DPO, SimPO and KTO. This is particularly noteworthy as DFT achieves these results in a single training stage, directly fine-tuning the pretrained Mistral-7B-v0.1 model without the preference data.

**AlpacaEval2 Results.** Finally, we briefly discuss the GPT4-as-a-judge evaluation on instruction following by reporting the length-controlled winning rate (LC) on AlpacaEval2. The comparison between DFT(2) with SFT and PO-based methods on self-play data is shown in Figure 2a, which demonstrates DFT(2) outperforms these baselines. The comparison between DFT(2) and SFT→PO approaches on preference data is shown in Appendix B.1, which shows that DFT(2) is competitive with some PO approaches using the preference data, such as KTO, but worse than SimPO and DPO. However, DFT(2) have much shorter lengths for the outputs with an average length of 1359. In contrast, KTO, DPO, SimPO have average lengths of 1449, 1477, 1868, respectively. The GPT4-as-a-judge evaluation tends to favor outputs that are longer. Nevertheless, DFT has competitive if not better performance on verifiable instruction following benchmark IFEval (cf. Table 3).

## 7.2 Ablation Studies

We present more results to illustrate the effectiveness of DFT compared with SFT and PO-based objectives, the advantage of our optimization using moving-average estimators $u$, and the effect of the number of generated samples $B$ in each

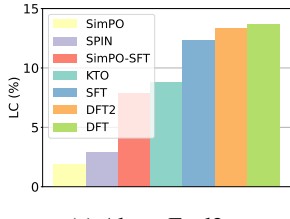

(a) AlpacaEval2.

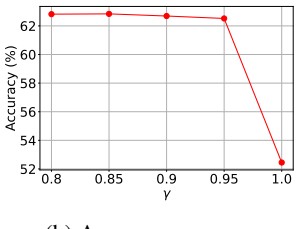

(b) Average accuracy.

Figure 2: (a) AlpacaEval2 LC win rate of DFT, SFT, and PO methods based on self-play data. (b) Average testing accuracy of DFT under different $\gamma$ values.

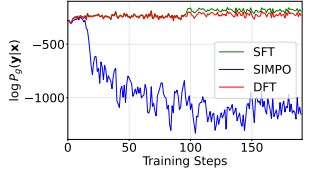

(a) Log-likelihoods of positives

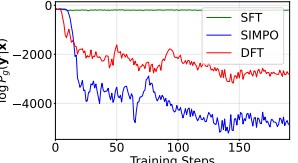

(b) Log-likelihoods of negatives

Figure 3: (a) Log-likelihoods of positive examples during training for different methods. (b) Log-likelihoods of negative examples during training for different methods.

iteration.

**Training Curves for the Log-likelihood.** Figure 3a and Figure 3b illustrate the learning dynamics of different methods by tracking the log-likelihood of positive and generated negative examples during training. We compare DFT with SFT and SimPO as in the Table 2. For positive examples (Figure 3a), DFT maintains a trajectory similar to SFT, while SimPO shows a decreasing trend, which means the objective of SimPO is not effective if using the self-generated negative examples for PO. For self-generated examples (Figure 3b), DFT successfully decreases the log-likelihood, demonstrating its effectiveness in distinguishing between positive and negative examples.

**Comparison with SFT→PO without Preference data** Table 4 presents results comparing DFT methods with SFT→PO without human preference data, where all methods use the same generated outputs from the base model as negative examples. We use the same UF→UF pipeline as in Table 3, while the only difference is that we replace the losing responses with the same generated responses as DFT in the PO stage of SFT→PO methods. These results demonstrate that the single stage training of DFT(2) is more effective than the two-stage training of SFT→PO methods for using the self-generated data.

**The advantage of using moving-average estimators $u$.** In the second training setting, we train DFT using different values of $\gamma$ ranging from 0.8 to 1.0. The value of 1.0 corresponds to using the mini-batch estimator of $g_i(\theta)$ for estimating the gradient. The average testing accuracy on

Table 2: Comparison between DFT, SFT, and PO methods on self-play data in the second training setting. All methods use the same UF winning responses as positive examples and the same generated outputs from the base model as negative examples, ensuring a fair comparison.

| METHOD | MMLU | TRUTHFULQA | HELLASWAG | WINOGRANDE | GSM8K | ARC | IFEVAL | AVG. |
|---|---|---|---|---|---|---|---|---|
| SFT | 62.18 | 50.04 | 83.59 | 78.06 | 45.26 | 63.65 | 49.72 | 61.79 |
| SPIN | 61.99 | 49.91 | 83.75 | 77.90 | 46.02 | 61.95 | 23.11 | 57.80 |
| SIMPO | **62.39** | 52.08 | 83.89 | 78.14 | 2.58 | 61.86 | 18.85 | 51.40 |
| SIMPO-SFT | 62.28 | 49.59 | 83.46 | 77.90 | 42.53 | 61.52 | 43.62 | 60.13 |
| KTO | 61.59 | 49.32 | 82.88 | **79.24** | 43.97 | 61.60 | 38.08 | 59.53 |
| ORPO | 62.26 | 48.26 | 83.07 | 79.16 | 45.41 | 62.20 | 53.41 | 61.97 |
| DPO-P | 62.01 | 48.66 | **84.03** | 78.61 | 40.48 | 62.20 | 25.32 | 57.33 |
| DFT | 61.69 | 52.23 | 83.95 | 78.37 | **48.22** | 64.25 | **51.20** | **62.84** |
| DFT2 | 61.66 | **54.14** | 83.20 | 77.82 | 45.49 | **64.42** | **51.20** | 62.56 |

Table 3: Comparison between DFT and SFT→PO approaches on preference data in the second training setting. DFT use only the UF winning responses, while SFT→PO methods use explicit preference pairs.

| METHOD | DATA | MMLU | TRUTHFULQA | HELLASWAG | WINOGRANDE | GSM8K | ARC | IFEVAL | AVG. |
|---|---|---|---|---|---|---|---|---|---|
| SFT→DPO | UC →UF | 57.49 | 53.15 | 83.60 | 77.43 | 30.55 | 61.52 | 39.93 | 57.67 |
| SFT→SIMPO | UC →UF | 58.33 | 50.67 | 83.39 | 76.95 | 33.36 | 61.86 | 40.48 | 57.86 |
| SFT→RRHF | UC →UF | 56.40 | 43.70 | 80.37 | 77.51 | 0.45 | 52.99 | 37.52 | 49.85 |
| SFT→R-DPO | UC →UF | 58.29 | 46.10 | 84.11 | 76.40 | 28.43 | 61.26 | 38.63 | 56.17 |
| SFT→CPO | UC →UF | 58.05 | 47.10 | 80.73 | 77.11 | 35.86 | 57.17 | 40.67 | 56.67 |
| SFT→IPO | UC →UF | 59.10 | 45.45 | 83.14 | 77.43 | 34.12 | 60.24 | 42.88 | 57.48 |
| SFT→KTO | UC →UF | 59.72 | 56.65 | 84.92 | 78.14 | 40.49 | 63.57 | 43.62 | 61.01 |
| SFT→KTO | UF →UF | 62.15 | 54.53 | 84.77 | 77.98 | 45.41 | 64.51 | 51.94 | 63.04 |
| SFT→DPO | UF →UF | 62.28 | 55.67 | 84.79 | 78.22 | 47.54 | 64.68 | 47.69 | 62.98 |
| SFT→SIMPO | UF →UF | 61.20 | 59.36 | 85.18 | 77.03 | 44.12 | 63.74 | 43.81 | 62.06 |
| DFT | UF $(\mathbf{x}, \mathbf{y}_w)$ | 61.69 | 52.23 | 83.95 | 78.37 | 48.22 | 64.25 | 51.20 | 62.84 |
| DFT2 | UF $(\mathbf{x}, \mathbf{y}_w)$ | 61.66 | 54.14 | 83.20 | 77.82 | 45.49 | 64.42 | 51.20 | 62.56 |

Table 4: Comparison of DFT with SFT→PO methods without human preference data, where all methods use the same generated outputs from the base model as negative examples.

| METHOD | MMLU | TRUTHFULQA | HELLASWAG | WINOGRANDE | GSM8K | ARC | IFEVAL | AVG. |
|---|---|---|---|---|---|---|---|---|
| DFT | 61.69 | 52.23 | 83.95 | 78.37 | 48.22 | 64.25 | 51.20 | **62.84** |
| DFT2 | 61.66 | 54.14 | 83.20 | 77.82 | 45.49 | 64.42 | 51.20 | **62.56** |
| SFT→DPO | 61.11 | 62.22 | 85.31 | 78.69 | 30.71 | 65.53 | 26.43 | 58.57 |
| SFT→SIMPO | 60.59 | 66.47 | 85.65 | 78.22 | 2.43 | 66.13 | 39.37 | 56.98 |

the seven benchmarks of Huggingface Open Leaderboard is shown in Figure 2b with more detailed results reported in Table 8 in Appendix B.2. It shows that using a moving-average estimator with $\gamma \in [0.80, 0.95]$ significantly improves the performance compared to not using the moving-average estimators corresponding to $\gamma = 1.0$, justifying the effectiveness of our optimization algorithm.

**The effect of $B$.** We compare different values of $B = 1, 2, 4$ in the second training setting on UF data. The results of DFT(2) and other PO-based approaches using the self-play data are shown in Appendix B.3. The results show that increasing $B$ from 1 to 2 improves the performance, especially on AlpacalEval2. However, further increasing

it to $B = 4$ decreases the performance. We suspect that this is probably due to overfitting, and expect more training data will accommodate a larger $B$, e.g., DFT on the larger MetaMathQA dataset with $B = 4$ is better than $B = 2$.

## 8 Conclusion

In this paper, we have proposed a discriminative probabilistic framework for finetuning a pretrained large language model without using any preference data or a reward model. Efficient and effective optimization algorithms are developed. Extensive experiments have demonstrated the effectiveness of the proposed methods compared with the standard supervised finetuning method and the existing preference optimization methods.

## Acknowledgments

S. Guo, V. Balmaseda, and T. Yang were partially supported by the NSF SCH grant 2306572.

## Impact Statement

This paper presents work whose goal is to advance the field of Machine Learning. There are many potential societal consequences of our work, none which we feel must be specifically highlighted here.

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

# A  Implementation Details

Our implementations are based on alignment-handbook[1] and TRL (von Werra et al., 2020) training framework. Below we provide comprehensive details about our experimental setup and implementation choices.

## A.1  Similarity Score

In all experiments, we use the unnormalized generative score $s_\theta(\mathbf{y}, \mathbf{x}) = \log P_g(\mathbf{y}|\mathbf{x})$ for DFT, and use the normalized generative score $s_\theta(\mathbf{y}, \mathbf{x}) = \frac{1}{|\mathbf{y}|} \log P_g(\mathbf{y}|\mathbf{x})$ for DFT2.

## A.2  Hyper-parameters Tuning

**Setting 1.** For both DFT and DFT2, we follow Yu et al. (2024), and set the batch size to 128, max sequence length to 512, and number of epochs to 3. We tune the learning rate in $\{5e-7, 8e-7, 2e-6\}$. For DFT, we tune the $\tau$ in $\{0.8, 0.9, 1.0\}$ and for DFT2, we tune the $\tau$ in $\{0.1, 0.2, 0.3\}$. Details of the chosen hyper-parameters are summarized in Table 5.

Table 5: Hyper-parameters for DFT methods under setting 1.

| Hyperparameters | DFT | DFT2 |
|---|---|---|
| $\tau$ | 1.0 | 0.1 |
| $\gamma$ | | 0.95 |
| Batch Size | | 128 |
| max sequence length | | 512 |
| Learning Rate | | 8e-7 |
| LR Scheduler | | Cosine |
| Warmup Ratio | | 0.1 |
| Optimizer | | AdamW |
| Epochs | | 3 |

**Setting 2.** We compare DFT variants against several baseline methods: SFT, SPIN, KTO, SimPO, and SimPO-SFT. For all methods, we use a batch size of 128, a maximum sequence length of 1024, and a training duration of 2 epochs. We perform learning rate tuning across $\{3e-7, 5e-7, 8e-7, 2e-6\}$, except for KTO where we use a larger learning rate of 5e-6. For DFT variants, we tune $\tau$ in $\{0.8, 0.9, 1.0\}$ for DFT and $\tau$ in $\{0.1, 0.2, 0.3, 0.4\}$ for DFT2, with $\gamma$ in $\{0.80, 0.85, 0.9, 0.95\}$ for both variants. The most effective values of $\tau$ are found to be 1.0 for DFT and 0.3 for DFT2. For the baseline methods, we tune their respective hyperparameters: $\beta$ in $\{0.01, 0.05, 0.1\}$ for SPIN; $\beta$ in $\{6, 8, 10, 12\}$ with a gamma-beta-ratio of 0.5 for SimPO; and $\beta$ in $\{6, 8, 10, 12\}$ with a gamma-beta-ratio of 0.5 and combining weight of 1 for SimPO-SFT. For KTO, we set $\lambda_U = 1$, and tune $\beta \in \{0.01, 0.05, 0.1\}$, $\lambda_D \in [B, 1.5B]$.

Details of the chosen hyperparameters are summarized in Table 6.

Table 6: Hyper-parameters for DFT methods under setting 2.

| Hyperparameters | DFT | DFT2 |
|---|---|---|
| $\tau$ | 1.0 | 0.3 |
| $\gamma$ | 0.85 | 0.90 |
| Batch Size | | 128 |
| max sequence length | | 1024 |
| Learning Rate | | 2e-6 |
| LR Scheduler | | Cosine |
| Warmup Ratio | | 0.1 |
| Optimizer | | AdamW |
| Epochs | | 2 |

---

[1] https://github.com/huggingface/alignment-handbook

## A.3    Implementation of SFT→PO

For SFT→PO methods under the UC→UF pipeline, we use the released checkpoints produced by SimPO (Meng et al., 2024), where models are first trained using SFT on UltraChat-200k for 1 epoch and then undergo preference optimization on UltraFeedback for 1 epoch. For SFT→PO methods under the UF→UF pipeline, we first train the SFT model on UltraFeedback with a learning rate of 2e-6 for 2 epochs, then conduct preference optimization on UltraFeedback for 1 epoch.

## A.4    Training Costs

We compare the training efficiency of different methods, with results shown in Figure 4. For DFT methods, the time is for training of 2 epochs. For SFT→SimPO and SFT→DPO, training time is split into two phases: an initial SFT phase of 2 epochs followed by a preference optimization phase of 1 epoch. Despite the higher computational cost compared to SFT alone, DFT2 offers comparable efficiency to the two-stage SFT→DPO pipeline. All experiments were conducted on 4×A100 80G GPUs. Generation time for negative samples (approximately 1.33 hours for both DFT methods) is not included in this comparison, as it can be performed offline as a preprocessing step.

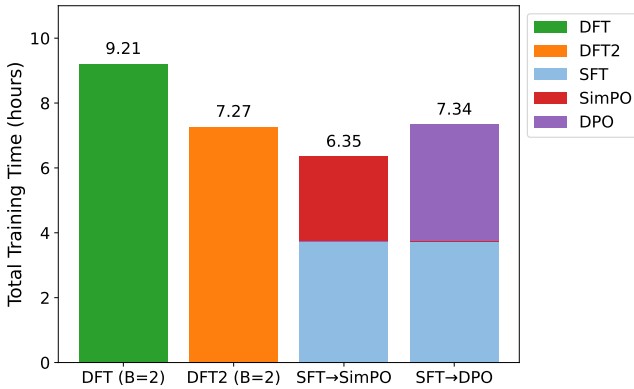

Figure 4: Comparison of total training time (in hours) for DFT methods and SFT→PO methods on the UltraFeedback dataset.

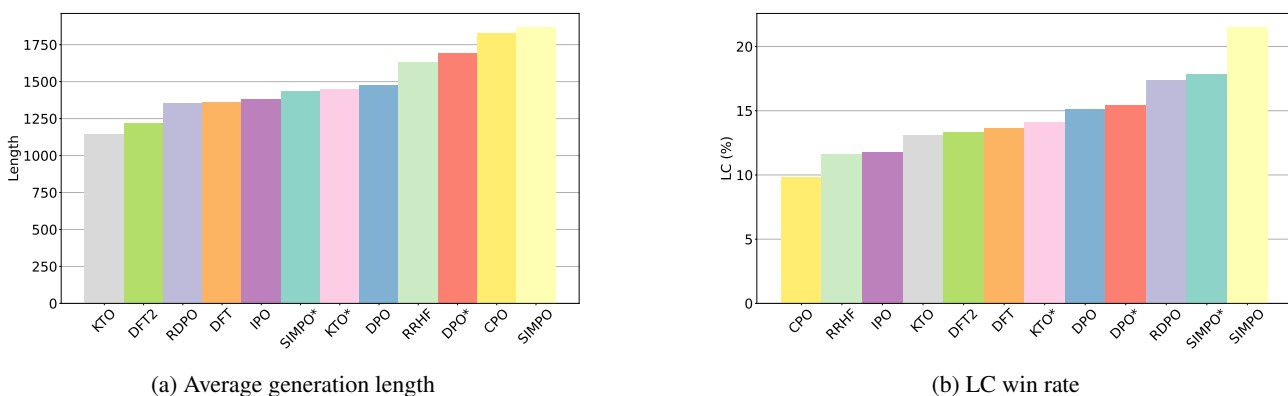

(a) Average generation length

(b) LC win rate

Figure 5: (a) AlpacaEval2 average generation length of DFT and SFT→PO approaches. (b) AlpacaEval2 LC win rate of DFT and SFT→PO approaches. SIMPO*, KTO*, and DPO* denote training under the UF→UF pipeline.

## A.5    Benchmark Details

Table 7 provides detailed information about the evaluation protocol used for each benchmark in the second training setting.

Table 7: Benchmark evaluation details including number of shots, metrics, and use of chat templates.

| Benchmark | Shot(s) | Metric | Chat Template |
|---|---|---|---|
| GSM8k | 5 | strict-match | ✗ |
| ARC | 25 | acc_norm | ✗ |
| HellaSwag | 10 | acc_norm | ✗ |
| TruthfulQA | 0 | acc | ✗ |
| MMLU | 5 | acc | ✗ |
| Winogrande | 5 | acc | ✗ |
| IFEval | 0 | prompt_level_strict | ✓ |

Table 8: Results of DFT with different $\gamma$ values for finetuning on UF winning data.

| METHOD | MMLU | TRUTHFULQA | HELLASWAG | WINOGRANDE | GSM8K | ARC | IFEVAL | AVG. |
|---|---|---|---|---|---|---|---|---|
| DFT w/ $\gamma = 0.8$ | 61.42 | 51.08 | 83.90 | 78.14 | 48.22 | 64.51 | 52.50 | 62.82 |
| DFT w/ $\gamma = 0.85$ | 61.69 | 52.23 | 83.95 | 78.37 | 48.22 | 64.25 | 51.20 | 62.84 |
| DFT w/ $\gamma = 0.9$ | 61.75 | 52.19 | 83.88 | 78.37 | 47.84 | 64.33 | 50.46 | 62.69 |
| DFT w/ $\gamma = 0.95$ | 61.82 | 50.78 | 83.93 | 78.30 | 46.47 | 64.42 | 51.94 | 62.52 |
| DFT w/ $\gamma = 1$ | 59.46 | 45.38 | 80.54 | 76.64 | 22.14 | 63.05 | 19.96 | 52.45 |

## B   Additional Evaluation Results

### B.1   AlpacaEval2 Results

Figure 5a and Figure 5b presents a detailed comparison between DFT variants and SFT→PO approaches on AlpacaEval2. The results demonstrate that DFT variants perform competitively to some PO methods such as SFT→KTO. They also tend to generate shorter output than models finetuned by most PO-based approaches. It is interesting to see that SFT→SimPO trained using UC→UF pipeline generates the longest outputs and it has the highest AlpacaEval2 score. However, its results on benchmark datasets in Table 3 is much worse than our method.

### B.2   DFT with Different $\gamma$

As shown in Table 8, $\gamma = 1.0$ leads to a significant performance degradation across all tasks, with particularly severe drops in GSM8K and IFEval. But the performance across this range $\gamma \in (0.8, 0.9)$ is consistently good.

### B.3   Detailed Analysis of $B$ Effect

We conducted extensive experiments varying the number of negative samples ($B$) used during training. Table 9 presents comprehensive results comparing DFT, SFT, and PO methods across different values of $B$ (1, 2, and 4). Figure 6 shows the AlpacaEval2 results. We can see that $B = 2$ has a dramatic improvement over $B = 1$, especially on AlpacaEval2 evaluation. However, increasing $B$ to 4 will decrease the performance.

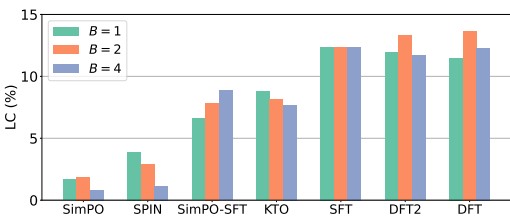

Figure 6: Comparison of different methods (DFT, SFT and PO methods) on AlpacaEval2 LC win rate across $B \in \{1, 2, 4\}$.

### B.4   Experiments with Different Base Models

To validate that our approach is effective across different model architectures, we conducted additional experiments using Qwen-2.5-0.5B and Llama3-8B-instruct models. For both models, we applied the same training protocol as Setting 2, using the UltraFeedback dataset for fine-tuning.

Table 9: Comparison between DFT, SFT and PO methods for fine-tuning Mistral-7B base model on self-play data across $B \in \{1, 2, 4\}$. The results show performance across 7 different benchmarks and their average.

| METHOD | MMLU | TRUTHFULQA | HELLASWAG | WINOGRANDE | GSM8K | ARC | IFEVAL | AVG. |
|---|---|---|---|---|---|---|---|---|
| SFT | 62.18 | 50.04 | 83.59 | 78.06 | 45.26 | 63.65 | 49.72 | 61.79 |
| $B = 1$ | | | | | | | | |
| SPIN | 62.16 | 50.23 | 83.67 | 78.06 | 46.10 | 62.03 | 19.59 | 57.41 |
| SIMPO | 62.29 | 50.75 | 83.84 | 78.06 | 2.88 | 61.69 | 19.41 | 51.27 |
| SIMPO-SFT | 62.53 | 48.83 | 83.55 | 77.82 | 43.44 | 61.60 | 42.70 | 60.07 |
| KTO | 61.27 | 50.28 | 83.30 | 78.53 | 40.79 | 63.74 | 42.51 | 60.06 |
| DFT | 61.76 | 50.89 | 83.95 | 77.98 | 46.70 | 64.42 | 50.65 | 62.33 |
| DFT2 | 61.92 | 52.87 | 83.02 | 78.14 | 44.81 | 64.68 | 51.57 | 62.43 |
| $B = 2$ | | | | | | | | |
| SPIN | 61.99 | 49.91 | 83.75 | 77.90 | 46.02 | 61.95 | 23.11 | 57.80 |
| SIMPO | 62.39 | 52.08 | 83.89 | 78.14 | 2.58 | 61.86 | 18.85 | 51.40 |
| SIMPO-SFT | 62.28 | 49.59 | 83.46 | 77.90 | 42.53 | 61.52 | 43.62 | 60.13 |
| KTO | 61.59 | 49.32 | 82.88 | 79.24 | 43.97 | 61.60 | 38.08 | 59.53 |
| DFT | 61.69 | 52.23 | 83.95 | 78.37 | 48.22 | 64.25 | 51.20 | 62.84 |
| DFT2 | 61.66 | 54.14 | 83.20 | 77.82 | 45.49 | 64.42 | 51.20 | 62.56 |
| $B = 4$ | | | | | | | | |
| SPIN | 61.94 | 49.60 | 83.70 | 77.98 | 46.02 | 62.03 | 20.70 | 57.42 |
| SIMPO | 62.31 | 51.39 | 83.83 | 77.90 | 4.47 | 61.86 | 19.04 | 51.54 |
| SIMPO-SFT | 62.46 | 49.47 | 83.47 | 78.06 | 42.15 | 61.69 | 39.93 | 59.60 |
| KTO | 60.99 | 49.87 | 82.06 | 78.61 | 41.62 | 61.86 | 40.11 | 59.30 |
| DFT | 61.52 | 51.80 | 83.97 | 78.53 | 46.32 | 64.51 | 50.65 | 62.47 |
| DFT2 | 61.78 | 53.67 | 83.35 | 77.90 | 45.64 | 65.02 | 48.80 | 62.31 |

Table 10 shows the performance of DFT and DFT2 compared to SFT when applied to the Qwen-2.5-0.5B model. Both DFT variants demonstrate improvements over standard SFT, with average gains of 0.44% and 0.54% respectively.

Table 11 presents the results when applying our methods to Llama3-8B-instruct. Both DFT variants show substantial improvements, achieving average gains of 1.88% and 1.86% respectively over SFT.

## B.5 Impact of Prompting Strategies

We investigated four different prompting strategies for generating the self-play data: (1) direct prompting without any special formatting; (2) structuring prompts using the same chat template as during fine-tuning; (3) in addition to (2), structuring prompts with beneficial system messages. (4) in addition to (2), structuring prompts with deliberately adversarial system messages. For setting (3), we follow the generation scripts of UltraFeedback. For setting (4), we add "You are an unhelpful assistant." to the system prompt. Examples of these prompts are given in the box at the end of the appendix. For all settings, we sample the negatives with the following parameters: a temperature of $0.7$, a top-p of $1.0$, a top-k of $50$, and a max-tokens of $320$.

Figure 7 illustrates the comparative performance of these strategies. Our analysis reveals that using structured prompts with adversarial system messages tends to generate more challenging negative samples, leading to better performance. Below, we provide an example of how each prompting strategy is implemented:

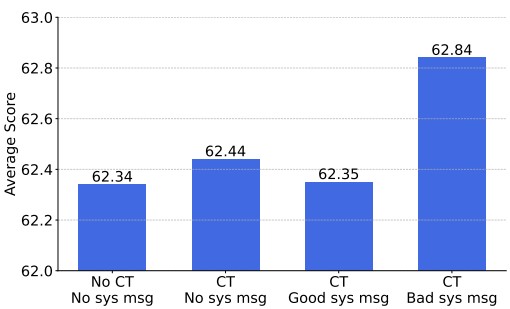

Figure 7: Average performance scores across different prompting strategies for generating negative samples. "CT" indicates whether chat template formatting was used, and "sys msg" refers to system messages. Results show that using chat templates with adversarial ("Bad") system messages achieves the best performance (62.84%), while beneficial ("Good") system messages yield a lower score (62.35%).

Table 10: Performance comparison using Qwen-2.5-0.5B as the base model

| METHOD | MMLU | TRUTHFULQA | HELLASWAG | WINOGRANDE | GSM8K | ARC | IFEVAL | AVG. |
|---|---|---|---|---|---|---|---|---|
| SFT | 47.34 | 42.88 | 51.20 | 55.41 | 33.43 | 36.60 | 17.56 | 40.63 |
| DFT | 47.49 | 42.77 | 51.30 | 56.59 | 35.56 | 36.43 | 17.38 | **41.07** |
| DFT2 | 47.15 | 44.86 | 51.57 | 56.67 | 32.83 | 37.37 | 17.74 | **41.17** |

Table 11: Performance comparison using Llama3-8B-instruct as the base model

| METHOD | MMLU | TRUTHFULQA | HELLASWAG | WINOGRANDE | GSM8K | ARC | IFEVAL | AVG. |
|---|---|---|---|---|---|---|---|---|
| SFT | 65.66 | 49.93 | 78.90 | 76.40 | 73.76 | 58.95 | 69.31 | 67.56 |
| DFT | 65.72 | 54.43 | 79.66 | 75.84 | 75.74 | 63.73 | 70.97 | **69.44** |
| DFT2 | 65.40 | 56.03 | 78.96 | 75.45 | 74.60 | 63.82 | 71.71 | **69.42** |

---

**Examples of Different Prompting Strategies**

**Raw Prompt Message:**

```
{"content": "Which animal has two hands, a hyrax or a dog?", "role": "user"}
```

**Direct Prompting:**

```
Which animal has two hands, a hyrax or a dog?
```

**Chat Template:**

```
<|user|>
Which animal has two hands, a hyrax or a dog?
<|assistant|>
```

**Chat Template + Good System Messages:**

```
<|system|>
The assistant should answer truthfully and be faithful to factual knowledge
as well as given contexts, never making up any new facts that aren't true
or cannot be grounded in the instruction.
<|user|>
Which animal has two hands, a hyrax or a dog?
<|assistant|>
```

**Chat Template + Bad System Messages:**

```
<|system|>
You are an unhelpful assistant.
<|user|>
Which animal has two hands, a hyrax or a dog?
<|assistant|>
```

## B.6   Impact of Sampling Temperature

This subsection examines how different sampling temperatures when generating negative examples from the base model affects DFT's performance. Table 12 presents the results of DFT trained with negative samples generated at four different temperature values ranging from deterministic sampling (0) to high-temperature sampling (1.0). The results show that moderate temperatures (particularly 0.7) yield the best overall performance across benchmarks.

Table 12: Results of DFT with different temperaure values during sampling.

| SAMPLING TEMPERATURE | MMLU | TRUTHFULQA | HELLASWAG | WINOGRANDE | GSM8K | ARC | IFEVAL | AVG. |
|---|---|---|---|---|---|---|---|---|
| 0 | 62.01 | 50.75 | 83.76 | 77.90 | 46.17 | 63.99 | 50.46 | 62.15 |
| 0.3 | 61.96 | 50.29 | 83.77 | 77.82 | 46.63 | 63.99 | 52.13 | 62.37 |
| 0.7 | 61.69 | 52.23 | 83.95 | 78.37 | 48.22 | 64.25 | 51.20 | 62.84 |
| 1.0 | 62.04 | 52.32 | 83.90 | 78.61 | 45.94 | 64.25 | 51.76 | 62.69 |

