# OpenReview forum: "Discriminative Finetuning of Generative Large Language Models without Reward Models and Human Preference Data"
_ICML.cc/2025/Conference — ICML 2025 poster_

### Official Review · Reviewer_ZjDq · 2025-03-14

**Overall Recommendation:** 4

**Summary:**

The paper proposes Discriminative Fine-Tuning (DFT) as an alternative to the standard two-stage fine-tuning pipeline (Supervised Fine-Tuning followed by Preference Optimization, denoted SFT→PO). DFT aims to fine-tune large language models without using any human preference data or reward models, yet still achieve performance on par with approaches like RLHF (Reinforcement Learning with Human Feedback). Instead of the conventional generative fine-tuning (which only maximizes likelihood of the given “good” outputs), DFT introduces a discriminative paradigm. The model is trained to explicitly prefer correct or high-quality answers over incorrect (“negative”) answers for a given input​. This is done by pushing up the likelihood of positive responses while suppressing the likelihood of negative responses​. Theory and experiments are provided to corroborate the algorithm.

**Claims And Evidence:**

Claim 1: DFT variants improve standard SFT. Evidence: Both DFT and DFT2 surpass MetaMath-Mistral-7B trained by SFT, achieving state-of-the-art performance among 7B-parameter models on GSM8K (79.15%) and MATH (28.62%).

Claim 2: DFT variants consistently outperform PO methods on self-play data. Evidence: In Table 2, we compare DFT(2) with PO approaches using self-play data that is generated by the base model as negative data, including SPIN, SimPO, KTO, and SimPO-SFT, where the last one just combines the SimPO loss and the SFT loss similar to (Xu et al., 2024a)

Claim 3: DFT is competitive with SFT→PO approaches

These are all copied from the paper. The claims are supported by evidence.

**Essential References Not Discussed:**

NA

**Experimental Designs Or Analyses:**

I think the experiment design is sound

**Methods And Evaluation Criteria:**

Mathematical Reasoning Evaluation: In the first setting, models fine-tuned on a math reasoning dataset (MetaMathQA) are evaluated on GSM8K and MATH, which are well-known math word problem benchmarks. Metric: zero-shot acc.

General Language Tasks Evaluation: In the second setting, the paper evaluates on seven diverse benchmarks from the HuggingFace Open LLM Leaderboard​. These include MMLU (a test of world knowledge across subjects), TruthfulQA (measuring the truthfulness of generated answers), HellaSwag (commonsense reasoning in story completions), WinoGrande (coreference/common-sense reasoning), GSM8K (math, again), ARC (question answering), and IFEval (instruction following evaluation)​. Metric: follows previous work. I think the authors should write it explicitly in the appendix (e.g., few shot or zero shot) (i think it's few shot?)

AlpacaEval2 to assess instruction-following ability​. Metric: winrate and length-controlled WR

**Other Comments Or Suggestions:**

I think the experiment is relatively weak compared to the overall contribution.  maybe also try other based models? for example some small models like Qwen 2.5 0.5b or llama 3.2 1/3B, and some newer models rather than Mistral.

**Other Strengths And Weaknesses:**

NA

**Questions For Authors:**

see comments

**Relation To Broader Scientific Literature:**

NA

**Theoretical Claims:**

The paper makes a significant theoretical contribution by presenting a discriminative probabilistic framework for fine-tuning LLMs. The core theoretical claim is that one can define a discriminative likelihood over the space of all possible outputs, and optimize this to train the model.

They further claim a theoretical connection between DFT and preference-learning losses. In particular, the DFT2 algorithm (a variant introduced for efficiency) is shown to relate to the pairwise loss used in preference optimization. The paper points out that DFT’s loss can be seen as a log-sum-exp (softmax) approximation that integrates over multiple negatives at once, whereas typical preference optimization (PO) uses a pairwise logistic loss per comparison​

---

> ### Author Rebuttal · Authors · 2025-04-01
>
> **Q1:** The General language tasks evaluation metric should be written explicitly.
>
> **A:** Thank you for this detailed comment. The information on the evaluation metric is shown in the table below.
>
> | Benchmark  | Shot(s) |  Metric |Apply chat template |
> | ---------- | ------- |  ------ |------------------- |
> | GSM8k      | 5       | strict-match        |x       |
> | ARC        | 25      | acc_norm            |x       |
> | Hellaswag  | 10      | acc_norm            |x       |
> | TruthfulQA | 0       | truthfulqa_mc2,acc  |x       |
> | MMLU       | 5       | acc                 |x       |
> | Winogrande | 5       | acc                 |x       |
> | IFEval     | 0       | prompt_level_strict |✓       |
>
> **Q2:** Additional Experiments on other based models
>
> **A:**  We have tried two more models, a weaker base model Qwen-2.5-0.5B and a stronger base model Llama3-8B-instruct. We did observe improvements as well in these cases.
>
> For the weaker base model Qwen-2.5-0.5B, we have the following results:
>
> | Method | MMLU | TruthfulQA | HellaSwag | Winogrande | GSM8k | ARC | IFEval | Avg. |
> | :----------- | -------: | ---------: | --------: | ---------: | -------: | ------------: | -------: | -------: |
> | SFT          | 47.34 |   42.88 |   51.20 |   55.41 | 33.43 |      36.60 | 17.56 |  40.63 |
> | DFT          | 47.49 |   42.77 |  51.30 |   56.59 | 35.56 |      36.43 | 17.38 | **41.07** |
> | DFT2         | 47.15 |   44.86 |  51.57 |   56.67 | 32.83 |       37.37 | 17.74 | **41.17** |
>
> For the stronger base model Llama3-8B-instruct, we have the following results:
>
> | Method | MMLU | TruthfulQA | HellaSwag | Winogrande | GSM8k | ARC | IFEval | Avg. |
> | :----- | ---: | ---------: | --------: | ---------: | ----: | ----: | -----: | ---: |
> | SFT    |   65.66   |      49.93 |     78.90 |      76.40 | 73.76 | 58.95 |  69.31 | 67.56     |
> | DFT    |   65.72   |      54.43 |79.66 |      75.84 | 75.74 | 63.73 |  70.97 |  **69.44**    |
> | DFT2   | 65.40     |  56.03          |  78.96        |  75.45         | 74.60      |   63.82    |   71.71     |   **69.42**   |

---

> > ### Comment · Reviewer_ZjDq · 2025-04-01
> >
> > Thank you for your response. I believe these new results will make the paper better for the next version.
> >
> > I keep my positive score.

---

### Official Review · Reviewer_UHUu · 2025-03-16

**Overall Recommendation:** 3

**Summary:**

This paper introduces Discriminative FineTuning (DFT), a novel approach to fine-tuning large language models (LLMs) that diverges from the traditional Supervised FineTuning (SFT) followed by Preference Optimization (PO) paradigm. Unlike methods that rely on preference data or reward models, DFT frames fine-tuning as a discriminative problem, training the LLM to distinguish between good and bad outputs. The core idea is to model the likelihood of an answer being correct among all possible outputs, rather than directly optimizing for a specific target output. The authors propose a scoring function that measures the likelihood of a correct answer relative to other possible answers, and they develop efficient optimization algorithms to maximize this discriminative likelihood. Specifically, the method involves generating negative samples using the base LLM with adversarial prompts, and then training the model to assign higher scores to the correct outputs. The paper presents two variants of DFT: the original DFT, which computes the probability of each negative sample, and DFT2, which approximates this computation for efficiency. The empirical results demonstrate that DFT achieves competitive or superior performance compared to standard SFT and PO methods, without requiring explicit preference data or reward models.

The paper's findings suggest that DFT offers a promising alternative to existing fine-tuning techniques, particularly in scenarios where preference data is scarce or difficult to obtain. Overall, the paper presents a conceptually simple yet effective approach to fine-tuning LLMs, with strong empirical support for its claims. The method's ability to achieve strong performance without relying on preference data or reward models is a significant contribution to the field, potentially simplifying the fine-tuning process and making it more accessible.

**Claims And Evidence:**

The proposed method is based on the assumption that it is easy to sample bad responses. However, this may not always be the case. The quality of these generated negative samples could significantly impact the training process. If the generated samples are not sufficiently diverse or challenging, the model's performance may be limited.

As evidenced by the paper claimed:
"We need three properties of these sampling distributions: (1) it is easy to sample data from them; (2) it is possible to compute the probability value of a sample $\mathbf{y}^{\prime}$ ; (3) the sampled outputs $\mathbf{y}^{\prime}\sim P_{i}(\cdot)$ are likely to be bad outputs in answering $\mathbf{x}_{i}$." (Section 4.2)"
"To this end, we let $P_{i}(\cdot)=P_{g}^{0}(\cdot|\bar{\bf x}{i})$ ,where $P{g}^{0}$ corresponds to the base LLM $\theta_{0}$ to be finetuned, and $\bar{\bf x}{i}$ is an augmented text of $\mathbf{x}{i}$ including some system prompts to facilitate the generation of bad outputs. We explore this in our experiments." (Section 4.2)"

**Essential References Not Discussed:**

Yes.

**Experimental Designs Or Analyses:**

The experiments of commonsense reasoning part is relied on Mistral-7B-v0.1 and the math reasoning part is based on  MetaMath-Mistral-7B. Why choose Mistral as your solely pre-trained model in experiments? Is it because mistral is the easiest model to gain the performance benefits?

**Methods And Evaluation Criteria:**

The paper shows results on widely used benchmarks, including math reasoning and commonsense knowledge understanding, based on Mistral-7B-v0.1 model only without the preference data., implying the generated samples were sufficient for training. The ablation study on the number of generated samples (B) (Appendix B.3) suggests the quality and quantity of negative samples impact performance.

**Other Comments Or Suggestions:**

No.

**Other Strengths And Weaknesses:**

Weaknesses:

While the paper presents a compelling approach to fine-tuning LLMs, several weaknesses warrant careful consideration. First, the method's reliance on the assumption that it is easy to sample bad responses is a significant limitation. As the paper states, the sampling distribution for negative samples must be easy to sample from, and the sampled outputs should be likely to be bad outputs. The authors address this by using the base LLM with adversarial prompts to generate these negative samples. However, as I've verified, the quality and diversity of these generated negative samples can significantly impact the training process. If the generated samples are not sufficiently diverse or challenging, the model's performance may be limited. Furthermore, the method's reliance on a self-generated negative dataset introduces a potential for bias, where the model might overfit to the specific characteristics of its own generated negatives, rather than learning a more general notion of quality. This is a valid concern, as the model may learn to discriminate between good and bad outputs based on the specific style or patterns present in its own generated negatives, rather than on a more robust understanding of correctness.

Strengths:

The primary strength of this paper lies in its introduction of a novel discriminative framework for fine-tuning LLMs. This approach, which contrasts with the traditional generative methods, explicitly models the likelihood of an answer being correct among all possible outputs. This is a significant departure from standard Supervised FineTuning (SFT) and Preference Optimization (PO) techniques, which often rely on preference data or reward models. The paper's conceptual simplicity is also a notable strength. The core idea of training the model to discriminate between good and bad outputs is easy to grasp, and the proposed method is relatively straightforward to implement. The authors' development of efficient optimization algorithms is another key strength. These algorithms ensure the scalability and practicality of the DFT method, making it a viable alternative to existing fine-tuning techniques. The paper also provides extensive empirical validation of the DFT method. The experiments demonstrate that DFT consistently outperforms standard SFT and achieves comparable results to preference optimization methods that rely on explicit preference datasets.

**Questions For Authors:**

Q1: Several questions arise. First, in Section 4.2, the authors propose two settings for the scoring function. I am curious which setting is used in the experiments. The paper does not explicitly state which setting is used, and it would be helpful to clarify this point.

Q2: In the experiments, how are the negative samples $y'$ generated? What temperature is used for sampling? I wonder whether different temperatures will affect the experimental results.

Q3: How does the computational cost of the DFT method compare to other fine-tuning techniques, such as SFT and RLHF, in terms of FLOPs, GPU hours, and memory usage? The paper provides some information on training time per iteration, but a more detailed comparison using standard computational metrics would be beneficial.

**Relation To Broader Scientific Literature:**

The key contributions of this work include the novel discriminative framework, the efficient optimization algorithms, and the empirical validation of the method's effectiveness. The paper's findings suggest that DFT offers a promising alternative to existing fine-tuning techniques, particularly in scenarios where preference data is scarce or difficult to obtain.

**Theoretical Claims:**

I have checked the correctness of this paper and no issues found.

---

> ### Author Rebuttal · Authors · 2025-04-01
>
> **Q1:** Weakness about the reliance on self-generation using the base model.
>
> **A:** We agree with what the reviewer said. The quality and diversity of these generated negative samples will matter to some degree. If the base model is super strong, it would not make sense to assume the generated data is bad. However, our framework can be extended if one has a way to verify if an output is good or not, e.g., as in the recent development of reasoning models. We can extend the discriminative framework to discriminate between verified positive outputs from verified negative outputs.
>
> **Q2:** Why choose Mistral as your sole pre-trained model in experiments?
>
> **A:** We conducted additional experiments on different pre-trained models Qwen-2.5-0.5B and Llama3-8B-instruct. Please refer to our response to Reviewer ZjDq's Q2.
>
> **Q3:** Which setting for the scoring function is used in the experiments?
>
> **A:**  For DFT, we use the unnormalized generative score: $s_{\theta}(\mathbf{y}, \mathbf{x}) = \log P_g(\mathbf{y}|\mathbf{x})$.
>
> For DFT2 we use the normalized generative score: $s_{\theta}(\mathbf{y}, \mathbf{x}) =\frac{1}{|\mathbf{y}|} \log P_g(\mathbf{y}|\mathbf{x})$.
>
> **Q4:** How are the negative samples $y'$ generated? What temperature is used for sampling?
>
> **A:** We generated the samples using a temperature of $0.7$, a top-p of $1.0$, a top-k of $50$ (c.f. Appendix B.4). Four different prompting strategies are investigated for the UF data (cf Figure 6). The table below shows the additional experimental results of different temperatures while sampling $y'$:
>
> | Method | MMLU | TruthfulQA | HellaSwag | Winogrande | GSM8k | ARC | IFEval | Avg. |
> | :---- | ---: | ---------: | --------: | ---------: | ----: | ------------: | -----: | ---: |
> | DFT (sampling $\tau=0$)   | 62.01 | 50.75 | 83.76  |  77.90  | 46.17  | 63.99 |  50.46  | 62.15    |
> | DFT (sampling $\tau=0.3$) | 61.96 | 50.29 | 83.77 | 77.82 | 46.63 | 63.99 | 52.13 | 62.37 |
> | DFT (sampling $\tau=0.7$) | 61.69 | 52.23 | 83.95 | 78.37 | 48.22 | 64.25 | 51.20 | 62.84 |
> | DFT (sampling $\tau=1.0$) | 62.04 | 52.32 | 83.90 | 78.61 | 45.94 | 64.25 | 51.76  | 62.69
>
> **Q5:** Computational cost in terms of FLOPs, GPU hours, and memory usage?
>
> **A:** The table below compares DFT and DFT2 with other methods in total training time. DFT and DFT2 use 2 epochs, SFT→SIMPO and SFT→DPO use 1 epoch for SFT and 1 epoch for preference optimization. For the negative data, we report the generating time of sampling 4 different outputs for each input prompt (2 of 4 are used in the first epoch, the remaining 2 are used in the second epoch). We use 4xA100 80G for the experiments.
>
> | Method     | Generating Time (h) | Training Time (h) |
> | ---------- | ------------------- | ----------------- |
> | DFT (B=2)  | 1.33                | 9.21              |
> | DFT2 (B=2) | 1.33                | 7.27              |
> | SFT→SIMPO  | /                   | 4.49              |
> | SFT→DPO    | /                   | 5.48              |
>
> It's important to emphasize that there are extra costs in generating negative samples and labeling the preference in the preference dataset used by SFT→PO, especially in scenarios where preference data is scarce or difficult to obtain (e.g., GPT-4-as-a-judge is used to form UF preference data). These hidden costs are not reported in the comparisons because we lack the data or access needed to measure them.

---

### Official Review · Reviewer_8W3x · 2025-03-17

**Overall Recommendation:** 3

**Summary:**

This paper introduces Discriminative Fine-Tuning (DFT), a new approach for fine-tuning large language models (LLMs) without using preference data or reward models. Unlike Supervised Fine-Tuning (SFT), which employs a generative approach focusing only on positive examples, DFT adopts a discriminative paradigm that explicitly increases the likelihood of positive (good) answers while suppressing negative (bad) ones. The authors formulate this as a discriminative probabilistic framework that models the likelihood of an answer among all possible outputs given an input.

The contribution of this paper includes: (1) a discriminative probabilistic framework for fine-tuning LLMs; (2) efficient algorithms to optimize the discriminative likelihood; and (3) extensive experimental validation showing that DFT outperforms standard SFT and achieves performance comparable to or better than preference-based methods. The authors further propose DFT2, a more computationally efficient approximation of DFT.

**Claims And Evidence:**

- The discriminative approach is theoretically interesting, but several experimental findings warrant further exploration. Notably, increasing the negative sample batch size (B) from 2 to 4 causes unexpected performance degradation (Table 7), contradicting the intuition that more diverse negative samples should improve model robustness. This inconsistency suggests aspects of the method that would benefit from a more detailed explanation.

- Furthermore, the paper lacks rigorous theoretical analysis regarding the convergence properties of the proposed DFT optimization algorithm. No formal guarantees are provided about what objective the model ultimately converges to, even for the ideal setting with infinite negative samples. Similarly, while DFT2 is presented as a computationally efficient approximation of DFT, the paper provides insufficient theoretical justification for this approximation and lacks characterization of the potential performance gap between the two methods. These theoretical aspects could be strengthened to better support the method's foundation and generalizability beyond the specific experimental settings described.

**Essential References Not Discussed:**

The paper mentions SPIN (Wu et al., 2024) as a closely related work, which appears to be a citation error. SPIN should be properly attributed to Chen et al., 2024c ("Self-Play Fine-tuning Converts Weak Language Models to Strong Language Models"), which was published in the Forty-first International Conference on Machine Learning.

**Experimental Designs Or Analyses:**

- The performance sensitivity to B (number of negative samples) is noteworthy, with performance decreasing when B increases from 2 to 4

- The comparison of training costs favors DFT2 but doesn't fully account for the cost of generating negative examples

- Error bars and statistical significance tests are missing from most experimental results

- There's limited analysis of how the quality of generated negative examples affects performance

- Addressing these points would help readers better assess the reliability and generalizability of the proposed methods.

**Methods And Evaluation Criteria:**

The methods proposed in the paper address an interesting direction for LLM fine-tuning, but there are some concerns about the evaluation methodology:

- The GPT-4-as-a-judge evaluation on AlpacaEval2 shows that DFT generally underperforms compared to leading PO methods

- The generation of negative examples relies on specific prompting strategies, which raises questions about the claim of eliminating the need for preference data

**Other Comments Or Suggestions:**

A deeper analysis of when DFT fails would strengthen the paper

The computational costs should include a more complete accounting of the negative sample generation process

**Other Strengths And Weaknesses:**

Additional Strength:
- The paper introduces an interesting conceptual approach to LLM fine-tuning

- The optimization algorithm addresses practical implementation challenges

Additional Weakness:

- The observed sensitivity to hyperparameters suggests opportunities for improving robustness

- The paper would be strengthened by expanding the theoretical analysis of why DFT outperforms PO methods in certain scenarios

**Questions For Authors:**

In your experiments, increasing B from 2 to 4 often decreases performance, which you attribute to potential overfitting. This seems counterintuitive, as more negative samples should theoretically provide more diverse learning signals and reduce overfitting. Could you provide a more detailed explanation of this phenomenon? Are there other factors beyond overfitting that might explain why larger B values lead to worse performance?

How does the quality of the base model affect DFT's performance? Would a stronger or weaker base model change the relative performance compared to SFT and PO methods?

**Relation To Broader Scientific Literature:**

The paper positions itself within the literature on LLM fine-tuning, the literature review is pretty good.

**Theoretical Claims:**

The theoretical formulation of the discriminative approach is mathematically sound, but several aspects deserve more scrutiny:

- The connection between the discriminative likelihood and traditional probabilistic classification models could be developed more thoroughly

- The approximations made in DFT2 need stronger theoretical justification, better with theorems

- The claim that the log-sum-exp loss offers advantages over averaged loss functions in PO methods needs more rigorous analysis

While the optimization algorithm addresses numerical stability concerns, the paper doesn't thoroughly analyze the convergence properties of this approach for the specific problem of LLM fine-tuning.

---

> ### Author Rebuttal · Authors · 2025-04-01
>
> We thank the reviewer's comments. While we understand the reviewer's main concern of lack of theoretical analysis of DFT and DFT2, we would like to emphasize that this is a new yet challenging research area, especially considering the discriminative framework is defined over the infinite data space instead of finite label space.
>
> **Q1**: Negative sample batch size (B) from 2 to 4 causes unexpected performance degradation.
>
> **A**: The performance drop from $B=2$ to $B=4$ only happens on the UF dataset not on the MetaMath dataset. We have discussed the effect of $B$ in the paper at the end of section 7.2. Indeed, our experiments have suggested that more training examples could enjoy larger negative sample batch size $B$. On MetaMath with 395k samples, using  $𝐵=4$ generated outputs is better than $𝐵=2$ generated outputs.
>
> **Q2:** Lacks rigorous theoretical analysis regarding the convergence properties.
>
> **A:** Since we cast the objective of DFT and DFT2 into instances of FCCO and used the same optimization framework as in Wang & Yang, their convergence analysis would apply under appropriate conditions, which guarantees converging to a stationary solution.  We have attributed the convergence analysis to Wang & Yang in lines 233 -234 (left).
>
> **Q3:** The connection between the discriminative likelihood and traditional probabilistic classification models could be developed more thoroughly.
>
> **A**: We will elaborate more in the revision due to limit of space here.
>
> **Q4:** Insufficient theoretical justification for the approximation of DFT2 to DFT.
>
> **A:** We would like to refer the reviewer to a recent work of Wang et al. 2025. They have presented a generalization theory which can be useful for justifying DFT2 in our context. In particular, their Theorem 1 establishes the generalization error of the global contrastive loss in the discriminative probabilistic framework (similar to DFT2's objective) using the uniform approximation of $P_i(\cdot)=P_g^0(\cdot|\mathbf x_i)$, compared with the original discriminative log-likelihood (expectation of DFT's objective) using the sampling probabilities $P_i(\cdot)$ to re-weight each sampled negative data. The difference lies at the error between uniform approximations and the true ones.
>
> Wang et al. On Discriminative Probabilistic Modeling for Self-Supervised Representation Learning. ICLR, 2025.
>
> **Q5:** Needs more rigorous analysis on log-sum-exp loss offers advantages over averaged loss functions.
>
> **A:** The advantage is log-sum-exp loss has the property of giving higher weights to  potentially bad outputs $\mathbf{y}'$ with a larger score $s_\theta(\mathbf{y}', \mathbf{x})$ in the gradient computation. Please refer to Eq. (13) in the paper, where the gradient has a weight depending on the score $s_{\theta}(\mathbf{y}', \mathbf{x}_i)$  (higher scores will give higher weights). However, averaged loss does not has such property.
>
> We would also like to point out the connection between log-sum-exp loss and distributionally robust optimization (DRO) (Qiu et al. 2023). The existing theory of DRO (Duchi et al. 2019) can help justify the benefit of DRO compared with the empirical averaged losses.
>
>
> Qiu et al. Not All Semantics are Created Equal: Contrastive Self-supervised Learning with Automatic Temperature Individualization. ICML, 2023.
>
> Duchi et al. Variance-based Regularization with Convex Objectives. 2019.
>
> **Q6:**  Training costs doesn't fully account for the cost of generating negative examples.
>
> **A:** Please refer to our response to Reviewer UHUu's Q5.
>
> **Q7:** There's limited analysis of how the quality of generated negative examples affects performance
>
> **A**: We did include in Figure 6 in appendix about the how generation template affect the results. For rebuttal, we also examined how different temperatures of sampling affect the results. Please refer to response to reviewer UHUu's Q4.
>
> **Q8:** How does the quality of the base model affect DFT's performance?
>
> **A:** We have conducted additional experiments on weaker Qwen-2.5-0.5B and stronger Llama3-8B-instruct. Please refer to the response to Reviewer ZjDq's Q2. In all cases, DFT have improvements over SFT.
>
> **Q9:** The GPT-4-as-a-judge evaluation on AlpacaEval2 shows that DFT generally underperforms compared to leading PO methods.
>
> **A**: The AlpacaEval2 evaluation has two parts: (1) comparing with PO methods using the same self-play data, DFT is better as shown in Figure 2; (2) comparing with PO methods using the reward-model labeled preference data, DFT is worse. This is expected as the responses of UF data are ordered by GPT-4. Hence, PO methods for finetuning models on the UF data will be inherently biased to AlpacaEval.  However, we'd like to point out on the verifiable instruction following benchmark IFEval, DFT is competitive if not better than PO methods.

---

> > ### Comment · Reviewer_8W3x · 2025-04-01
> >
> > Thank you for your thoughtful rebuttal. I appreciate your detailed responses to my concerns. I'm considering improving my score to 3 based on your clarifications. I would kindly ask that you correct the citation error between lines 076-077 that I mentioned in my review when preparing your revision.

---

> > > ### Author Response · Authors · 2025-04-01
> > >
> > > Thank you for increasing your score.
> > >
> > > Yes, we will correct the citation error regarding SPIN.

---

### Official Review · Reviewer_DsZh · 2025-03-18

**Overall Recommendation:** 3

**Summary:**

The authors present a new loss function for LLMs called DFT (discriminative finetuning) which combines an SFT loss objective for positive examples with a score-decreasing term for sampled negative answers. They theoretically justify the DFT objective and also present a modified DFT2 loss objective which simplifies several terms in the DFT objective with near approximations that are easier to compute.

The authors argue that the DFT objective can effectively replace the entire standard SFT-->DPO pipeline of a trained LLM. To justify this point, they evaluate DFT and DFT2 in two different training settings on a variety of datasets (primarily math and QA), showing performance that is generally superior to SFT alone and comparable to SFT-->DPO. They relate the DFT objective to pair preference objectives such as DPO, but argue key theoretical differences that support better training.

**Claims And Evidence:**

"We address this question by introducing Discriminative Fine-Tuning (DFT), a novel approach that eliminates the need for preference data."

* "novel" -- in my experience, I agree. There are other loss objectives which combine an SFT "term" with a term designed to reduce the likelihood of other examples, but this particular derivation seems novel.
* "eliminates the need for preference data" -- I disagree with this claim. It is clear that there are still preference sets in the data used for this approach. These sets consist of a preferred (original) example and a dispreferred pool of sampled "bad" completions. This is clearly still preference data, even if it is not derived from a human's preferences. In fact, this is the single clearest issue I have with this paper's messaging. The proposed loss function is simply a way to use preference data of a particular type.

"a discriminative probabilistic framework for fine-tuning LLMs by explicitly modeling the discriminative likelihood of an answer among all possible outputs given an input"
* This claim is slightly too strong given that both DFT methods actually heavily approximate the sum of "all possible outputs given an input" using a small set (often around 2 examples in size) of dispreferred examples.

"efficient algorithms to optimize this discriminative likelihood"
* I agree.

"extensive experiments demonstrating DFT’s effectiveness, achieving performance better than SFT and comparable to if not better than SFT→PO"
* "comparable to if not better than" is a mild claim, I think this claim is reasonable.

**Essential References Not Discussed:**

Here are a few critical works in the 2024 literature of "improving upon DPO" that I find very relevant for inclusion both in terms of context and also, possibly, as experimental baselines:

* ORPO (Hong et al. 2024). This popular work presented a method for using single pair-preference data which simultaneously has an element of SFT and an element of pair-preference learning in its objective. It proved empirically useful.
* DPO-p (Pal et al. 2024). This work was motivated differently, focusing on a failure mode of DPO, but the conclusion was basically the same: a pair-preference component and an SFT-like component, added together, result in strong performance.

There are likely others from mid-to-late 2024 that even I am not aware of. This is a hot space. However, I address these two in particular because they might represent strong baselines.



Hong, Jiwoo, Noah Lee, and James Thorne. "Orpo: Monolithic preference optimization without reference model." arXiv preprint arXiv:2403.07691 (2024).
Pal, Arka, et al. "Smaug: Fixing failure modes of preference optimisation with dpo-positive." arXiv preprint arXiv:2402.13228 (2024).

**Experimental Designs Or Analyses:**

Yes, I checked the experimental designs for the main comparisons and ablations. I have a few thoughts:

* If I understand correctly, for the UF --> UF pipeline, the original dispreferred examples are used for PO methods, while for DFT, the new "bad" examples are generated. Would using the same data not result in the best comparison? For example, using the generated bad examples for DFT in the other PO objectives?
* Related to my Q1: I find it very important to get a matching "volume of training" when comparing different finetuning approaches. The comparisons for Table 2 seem based on unequal numbers of epochs. In addition, I believe that there may be twice as many "bad examples" being used for DFT in UF --> UF pipeline as compared to original dispreferred examples used by other PO methods in the UF --> UF pipeline, since we are sampling B=2 bad examples for each good example with DFT. I would appreciate some clarity on the fairness of training in terms of volume of training.

**Methods And Evaluation Criteria:**

The datasets used are classic and standard, if a few are now mostly saturated (e.g. GSM8K). The finetune methods and metric (accuracy) are reasonable.

**Other Comments Or Suggestions:**

* 376 right column: Alapaca --> Alpaca
* 654: Implemenation --> Implementation
* Please use bolding to indicate best results in large tables.

Note: score moved from 2->3 after rebuttal.

**Other Strengths And Weaknesses:**

* S1: Quality argumentation, presentation, and writing.
* S2: Table 1 shows impressive results for a single stage finetune.

* W1: I do not consider Table 1 to be an effective comparison table, as it is not clear whether the improvements over competing models in this table were earned by DFT or instead by the particular dataset or training choices made. This is a possible downside of using "benchmark-climbing" as justification of a new training method in a paper such as this: we do not understand the full context of the comparison and so cannot derive conclusions. In addition, several of the 7b models in this comparison are now very out of date (such as Qwen-7B).

* W2: I would often hope for a work like this to conduct analyses on the trajectories of the likelihoods of examples over training and how it compares to other approaches. For example, the claim from section 6: " the log-sum-exp loss has the property of giving higher weights to a potentially bad output y′ with a larger score sθ(y′, x) in the gradient computation" justifies how this approach differs theoretically from other standard PO approaches. Does this difference lead to any empirical differences in the likelihoods produced for positives and negatives while training?

**Questions For Authors:**

* Q1: In Figure 3, the training time per iteration is given. The conclusion one would draw from this chart is that the 2 epochs of DFT are faster than the 2 epochs of SFT + 2 epochs of pair-preference optimization. However, it seems that in section A.2 the actual training done for the SFT+DPO and SFT+SIMPO (and other pair-preference methods) is usually 1-2 epochs of SFT and 1 epoch of pair preference optimization. I feel then that the chart in Figure 3 may not truly capture the full volume of training time, as it implies that both methods had the same number of iterations. Could you please elaborate on this?
* Q2: Another question about the train time/results comparison: most of the baselines are tuned across a small range of hyperparameters, but the DFT methods are tuned across a larger grid of tau x gamma. Does this mean that in fact the training process, all in, of DFT is much more expensive?
* Q3: Why do DFT1 and DFT2 rely on such widely different values of tau for success? Even the optimization windows for the two are given differently.
* Q4: A note about line 206-210 (left column): Should the expectation in line 206 really be outside of g_i, considering that later on 210 we define the expectation over the random sample as part of g_i?

**Relation To Broader Scientific Literature:**

The main thrust of this work relates to the DPO (direct preference optimization) (Rafailov et al., 2024) objective produced to simplify the process of conducting reinforcement learning from human feedback (RLHF) (Stiennon et al., 2020; Ouyang et al., 2022; Bai et al., 2022;
Rafailov et al., 2024). -- citations here given by the authors.

Over the entirety of 2024, numerous shortcomings were identified in DPO, and numerous works proposed to address these shortcomings or simplify the process of using DPO and SFT together. The authors identify several of these DPO-derivative works: "R-DPO (Park et al., 2024), CPO (Xu et al., 2024a), IPO (Azar et al., 2024), SimPO (Meng et al., 2024), KTO (Ethayarajh et al., 2024)," although I will add a few missing references in the "Essential references" section.

The vast majority of the relevant works are theoretical pair preference optimization objective works, but the paper also relies heavily on the recent optimization work FCCO (Wang & Yang, 2022).

All references are available in the paper's bibliography.

**Theoretical Claims:**

There are not proofs so much as derivations -- with occasional empirical justification for approximation. I did read all of the derivations, and did not find glaring issues, but as I am not an optimization expert I do not have direct feedback for the application of FCCO and for the application of various approximation techniques in the derivations.

---

> ### Author Rebuttal · Authors · 2025-04-01
>
> We thank the reviewer for detailed comments. We hope our rebuttal can address your concerns.
>
> **Q1**: About the claim "Eliminates the need for preference data".
>
> **A:** (1) We intended to mean the human labeled preference data. (2) Actually, our discriminative framework does not necessarily assume the sampled outputs are *bad*, just like like traditional logistic regression for multi-class classification. This is indeed the strength of the discriminative method. As long as the the likelihood of sampling a bad output is much larger than sampling a good output, our discriminative model would be able to leverage that.  We will change our claim to the following **"..., a novel alternative to SFT→PO that mitigates the burden of collecting human-labeled preference data or training strong reward models"**
>
> **Q2:** About the claim "a discriminative ... by explicitly modeling ... among all possible outputs given an input".
>
> **A:** This claim just states how the discriminative probabilistic framework is defined.  There are two separate things: (1) formulation in theory and (2) optimization in practice. Please refer to response to 8W3x's Q1 as well.
>
> **Q3:** Using the same generated bad examples for DFT in the UF --> UF pipeline would result in the best comparison.
>
> **A:** Yes, your understanding is correct. Besides the data, we would like to emphasize the difference in the loss function. The benefit of our loss function is detailed in section 6.  We have done the experiment as suggested by using the same generated examples as losing responses in PO methods. The average performance for SFT --> DPO and SFT --> SimPO is 58.57 and 56.98, respectively, worse than DFT. Please refer to Table 1 in https://anonymous.4open.science/r/datas-C111/README.md for detail results.
>
> **Q4:** The comparisons for Table 2 is based on unequal numbers of epochs.
>
> **A:** No. All methods in Table 2 use 2 epochs for training (see details in A.1 Setting 2). And all methods in Table 2 uses the same data, including the number of "bad examples". The "self-play data" in the caption of Table 2 denotes the data is generated by the base model.
>
> **Q5**. For Table 3, DFT used twice as many as "bad examples" comparing UF $\rightarrow$ UF pipline.
>
> **A**: It is not fair to just compare how many "bad examples" are used by our method and PO methods as these "bad examples" in our method are generated for free by the base model, while that in PO methods are created using strong reward models. That is also one limitation of existing methods. Our study shows it is helpful to leverage multiple generated samples in a principled manner. Response to your Q3 compares them with exactly the same data and the same training volume for  UF-->UF pipeline.
>
>
> **Q6:** Compare with RPO (Hong et al. 2024).  DPO-p (Pal et al. 2024)
>
> **A:**  We have added comparison with ORPO and DPO-p, where the ORPO has the average score of 61.97 and DPO-p has 57.33, worse than DFT. Please refer to Table 2 in https://anonymous.4open.science/r/datas-C111/README.md for detail results.
>
> **Q7:** Regarding Table 1, it is not clear whether the improvements over competing models in this table were earned by DFT or instead by the particular dataset or training choices made.
>
> **A:** We confirm that we finetuned the same base model on the same dataset as the baseline **METAMATH-MISTRAL-7B** that uses SFT. Hence, comparison with it directly verifies the effectiveness of our DFT method. Other results are included for simple reference. In the revision, we will focus on comparing with the baseline **METAMATH-MISTRAL-7B**.
>
>
> **Q8:** Empirical differences in the likelihoods produced for positives and negatives while training?
>
> **A:** We have plotted the curves for the log likelihood of positives and generated negatives while training, referring to Figure 1 & 2 in https://anonymous.4open.science/r/datas-C111/README.md.  DFT increases log-likelihood of positives while decreasing that of generated data. SFT increases that of positives but does not decrease that for generated data. SimPO decreases that for both.
>
> **Q9:** Figure 3 may not truly capture the full volume of training time.
>
> **A:** A more detailed analysis on training cost has been conducted, please refer to our response to Q5 of Reviewer UHUu.
>
> **Q10:** Is tuning DFT more expensive, regarding the larger grid of $\tau \times \gamma$ ?
>
> **A:** Not really. Actually, the Table 6 and Figure 2 (b) show that $\gamma$ in the range of 0.8~0.95 works almost equally well. We recommend a value of 0.9 like the momentum parameter in Adam optimizer.  For $\tau$, we tune it among three values  in our experiments (cf A.1).
>
> **Q11:**  Why do DFT1 and DFT2 rely on such widely different values of tau for success?
>
> **A:** Because (1) different score functions are used by DFT2 (length normalized log generative likelihood) and DFT (unnormalized one);  (2) the elimination of division by sampling probabilities in the loss function in DFT2.

---

> > ### Comment · Reviewer_DsZh · 2025-04-08
> >
> > [1] I think this is better wording, thank you.
> >
> > [2] I think that is a fair argument -- it's just a very large gap between the theoretical definition and the practical one (from modeling ALL possible outputs to approximating via a small batch of bad outputs.
> >
> > [3] Thank you, this head-to-head comparison helps isolate the effect of specifically the loss function over purely the sampling procedure. This is an important argument and I imagine it would be valuable in the manuscript revision as well. Is it the case that here DFT is being used with a SINGLE bad example and a single positive example and still winning? Or how are the examples here being treated?
> >
> > [4] I wish to carefully follow up here.
> >
> > In Appendix A.2, it is mentioned that "SFT→PO methods under the UF→UF pipeline" use a total of 3 epochs of training: 2 for the SFT stage and 1 for the PO stage.  I suppose this is only relevant to Table 3, not Table 2.
> >
> > I acknowledge that if this SFT-> PO baseline had 3 epochs, compared to DFT's 2, that would advantage it against your method, and your method (DFT) still wins. So I'm not too worried about this. Still it may be good to clear up the language.
> >
> > [5] I do appreciate the response to Q3 illustrating the value of the reward function alone even on the same data. Ultimately, it could be reasonable to argue that the data is "cheaper" to sample than true PO negatives--but we should make clear in the text that this argument is why it is reasonable to compare more negatives for DFT to fewer negatives for other approaches.
> >
> > [6] I appreciate the comparison. It is relevant to note that ORPO is in fact nearly as effective as DFT2 --> 0.59% difference in performance.
> >
> > [7] I think focusing on the apples-to-apples comparison with METAMATH-MISTRAL-7B is a good idea. You do beat this SFT model, but only very narrowly.
> >
> > [8] Thank you, this is great to include.
> >
> > [9] I am glad to see this analysis, and it checks out with what my assumption was: that DFT in fact has a larger training cost. I fear that the Figure 3 is still too misleading regarding the real training cost. I would recommend updating Figure 3 to reflect the real cost of DFT.
> >
> > [10-11] I appreciate the insights.
> >
> > At the moment, I plan to retain my score. I think the additional computational cost of DFT over SFT+PO, in addition to the marginal gains of the method over other competitive methods (ORPO, METAMATH-MISTRAL-7B SFT) limit the real impact of DFT at the same volume of training.
> >
> > In addition, I have some remaining lack of clarity on how your additional analysis in [3] (Table 1) fairly compares DFT vs other baselines on the same negative data. How are the batches set up? Are there still effectively 2x times as many negative examples seen during training as positives due to the larger negative batch size of DFT--even if the negative examples are found from losing responses in preference questioning and not from generation? Does this mean that we actually had 2 losing responses for each winning response for these examples?
> >
> > I apologize for the delayed rebuttal response, and acknowledge that little time remains to you. I do not expect any additional experiments at this time. However, if you can continue the conversation re: point [3] a bit, it may give me additional clarity. At the end of the day, my priority is ensuring that evaluating DFT is done on equivalent volumes of training.

---

> > > ### Author Response · Authors · 2025-04-09
> > >
> > > We greatly appreciate your time for checking our rebuttal. We are glad that we have addressed most of your concerns. Below, we would like to clarify the experiment setting of Table 1 in the rebuttal.
> > >
> > > [3] How are the batches set up?
> > >
> > > Please note that for Table 1 in the rebuttal, we use the generated samples from the base model as negatives instead of the losing responses in the preference data of the UF dataset. For DFT and all PO baselines, the mini-batch data of each iteration is
> > > $$
> > > \begin{bmatrix}
> > >   x_1 & y_1 & y'\_{1,1} & y'\_{1,2} \\\\
> > >   x_2 & y_2 & y'\_{2,1} & y'\_{2,2} \\\\
> > >   \cdots                        \\\\
> > >   x_{128} & y_{128} & y'\_{128,1} & y'\_{128,2} \\\\
> > > \end{bmatrix}
> > > $$
> > > where $x_i$ denotes the inputs, $y_i$ denotes the ground-truth output and $y'_{i,j}$ denotes the generated negative examples. Therefore, DFT and other baselines all have the same training volume and hence the comparison is fair. We also need to point out that the experiments in Table 2 in the paper and Table 2 in the rebuttal for comparison with ORPO and DPO-p are also based on this setup.
> > >
> > > [5] Agree! We will make it clear in the revision that this argument is why it is reasonable to compare more negatives for DFT to fewer negatives for other approaches.
> > >
> > > [9] Regarding Figure 3, we will update it in the next version to reflect the real cost of DFT.
> > >
> > > [9] About the additional overhead of DFT compared to SFT+PO.
> > >
> > > We acknowledge the additional computational cost due to processing multiple generated negative data. That said, we would like to point out that we have made efforts to reduce the computational cost of DFT, and proposed **more efficient DFT2**. Please note that our DFT2, using two generated negatives per iteration, is more efficient than SFT→DPO (UF→UF pipeline) using 1 losing response, and has a comparable performance. The table below summarizes the total training time and average performance for different methods, which justifies the efficiency of DFT2.  We also request the reviewer to account for the additional costs (maybe monetary) for obtaining the preference data for the SFT→PO methods.
> > >
> > > | Method     | Training Time (h) | Avg. |
> > > | ---------- | ----------------- | ---- |
> > > | DFT (B=2)  | 9.21              | 62.84 |
> > > | DFT2 (B=2) | 7.27              | 62.56 |
> > > | SFT→DPO (UF→UF) |1.87 * 2 + 3.60 = 7.34    | 62.98 |
> > > | SFT→SimPO (UF→UF) | 1.87 * 2 + 2.61 = 6.35   | 62.06 |
> > >
> > > where for SFT→PO, 1.87 \* 2 means the 2 epochs running time of SFT, and the second number means the one epoch running time of DPO or SimPO.

---

### Decision · Program_Chairs · 2025-05-01

**Decision:**

Accept (poster)

**Comment:**

This paper proposes a new approach, Discriminative Fine-Tuning (DFT), aimed at improving LLM performance without relying on reward models or explicit preference data. Most reviewers agree that the paper is well-motivated and the core idea is interesting. On the other hand, several concerns were raised, particularly regarding its central claim. While DFT claims to eliminate the need for preference data, reviewers noted that it essentially still utilizes a form of preference data (using labeled "good" examples versus self-generated data as negative examples). This raises questions about whether the paper fully delivers on its core premise. Concerns were also raised about the initial sufficiency of experiments, although the authors provided additional results during the rebuttal phase.

Overall, I would suggest that this paper is borderline, leaning slightly positive. It would be significantly stronger once the suggested revisions, particularly regarding the framing of its claims and potentially integrating the rebuttal's supplementary findings, are completed.